# Unintended Selection: Persistent Qualification Rate Disparities and Interventions

**Reilly Raab**
Computer Science and Engineering
University of California, Santa Cruz
Santa Cruz, CA 95064
reilly@ucsc.edu

**Yang Liu**
Computer Science and Engineering
University of California, Santa Cruz
Santa Cruz, CA 95064
yangliu@ucsc.edu

## Abstract

Realistically—and equitably—modeling the dynamics of group-level disparities in machine learning remains an open problem. In particular, we desire models that do *not* suppose inherent differences between artificial groups of people—but rather endogenize disparities by appeal to unequal *initial conditions* of insular subpopulations. In this paper, agents each have a real-valued feature $X$ (*e.g.*, credit score) informed by a "true" binary label $Y$ representing *qualification* (*e.g.*, for a loan). Each agent alternately (1) receives a binary classification label $\hat{Y}$ (*e.g.*, loan approval) from a Bayes-optimal machine learning classifier observing $X$ and (2) may update their qualification $Y$ by imitating successful *strategies* (*e.g.*, seek a raise) within an isolated group $G$ of agents to which they belong. We consider the disparity of qualification rates $\Pr(Y = 1)$ between different groups and how this disparity changes subject to a sequence of Bayes-optimal classifiers repeatedly retrained on the global population. We model the evolving qualification rates of each subpopulation (group) using the replicator equation, which derives from a class of imitation processes. We show that differences in qualification rates between subpopulations can persist indefinitely for a set of non-trivial equilibrium states due to uniformed classifier deployments, even when groups are identical in all aspects except initial qualification densities. We next simulate the effects of commonly proposed fairness interventions on this dynamical system along with a new feedback control mechanism capable of permanently eliminating group-level qualification rate disparities. We conclude by discussing the limitations of our model and findings and by outlining potential future work.

## 1 Introduction

Algorithmic prediction is increasingly used for socially consequential decisions and may determine individual access to information, education, employment, credit, housing, medical treatment, freedom from incarceration, or freedom from military targeting [1–5]. This situation raises technical challenges and ethical concerns, particularly regarding the dynamics of systemic inequalities and attendant harms to society [6–8]. Nonetheless, realistically—and equitably—modeling the dynamics of disparity in machine learning remains an open problem.

Research historically considered the fairness of algorithmic predictions in terms of statistical (in)consistencies [9] (*e.g.*, across groups [10–16] or between similar individuals [10, 11]), preference guarantees [17–19], or causal considerations [19, 20] but ignored the *response* of a population to new prediction policies. For instance, the proportions of potential loan applicants in each group that will seek higher wages, falsify income, or forego application might *change* if banks use new policies to approve or deny loans, possibly countering fair intent. We refer this class of fairness definitions as *normative present fairness*.

35th Conference on Neural Information Processing Systems (NeurIPS 2021).

Efforts to model such population response [21–31]and the autonomous dynamical systems arising from mutual recursion with myopically updating prediction policies [21–29] have intensified, but it has remained to plausibly explain persistent disparities under group-independent prediction policies—*i.e.*, those that do not discriminate on the basis of group membership—without assuming a setting that is structurally imbalanced between groups. Our paper contributes to these efforts and considers the long-term consequences of machine learning on inter-group disparities when a sequence of classifiers induces dynamics within the rates of strategy adoption in each group. Upon adopting a dynamical framework, we note that yet another operationalization of fairness arises: the asymptotic equality of latent variables (*i.e.*, those causally responsible for outcome disparities) between groups. This notion of *long-term fairness* need not be consistent with *normative present fairness*, which may actively combat it, highlighting a tension between *ends* and *means* for fairness considerations.

## 1.1 Our contributions

Herein, we describe an *equitable* model of population response: one which does not suppose inherent differences between groups of people but endogenizes disparities by appeal to unequal *initial conditions*, accounting for group-specific environmental conditions as dynamical variables. We reform our notion of "groups" (*i.e.*, *subpopulations*) to appeal to natural boundaries of information exchange rather than artificially imposed classes of people. We thus offer a potentially more meaningful way to group individuals in discussions of fairness, asserting that, when considering such networks of peer exchange, "sensitive attributes" such as race, sex, color, *etc.* might not correspond to meaningful divisions of people, which depend on social context. Finally, we recognize fairness interventions as dynamical control policies that (un)intentionally select the future trajectories of a given system. We therefore allow ourselves to consider interventions that explicitly incorporate *feedback* from dynamical variables—rather than relying on fixed, prescriptive modifications of predictor loss functions.

Our first contribution is to propose a model of *classifier-induced* group-level strategy adoption that is (1) equitable, *i.e.*, free from structurally asymmetric assumptions as described above, (2) capable of explaining persistent disparities under Bayes-optimal, group-independent policies, and (3) derivable from plausible, localized information exchange between individuals. Specifically, we appeal to the replicator equation, an established model for evolutionary phenomena without mutation, to model how competing *strategies for qualification* (which determine true machine learning labels $\{0, 1\}$, affecting agent utilities) replicate within groups (*i.e.*, isolated subpopulations that differ only in size and initial proportions of qualified individuals). We ground statements with a running example involving loan applications (elaborated upon in Section 2.2) for which qualification (label $Y = 1$), interpreted as being in the public interest, implies future repayment of a loan for an applicant with feature profile $X$. As we avoid assuming inherent differences between groups, we consider the label-conditioned feature distribution $\Pr(X \mid Y = y)$ as group-independent and define qualification disparity in terms of differences in group qualification rates $\Pr(Y = 1)$. We formulate our model in Section 2, emphasizing that only the *profile of strategies* in each subpopulation is subject to evolution—narrowly qualified by the competition between strategies for replicative success—rather than the subpopulations themselves. The persistence of disparity is thus attributed to classifier policy.

Our second contribution, in Section 3, is a rigorous examination of the dynamical system formed by the replicator equation and an updating, group-independent, Bayes-optimal classifier policy, including a characterization of its equilibrium states with linear stability analysis. We identify the set of stable interior states of the system as a stable hyperplane and show that any initial state with non-zero total qualification disparity, defined in Section 3, will continue to exhibit non-zero disparity asymptotically if the state attracts to the stable hyperplane (Theorem 10). In this sense, we claim that qualification rate disparity persists indefinitely for this setting.

Our final contribution, in Section 4 is to consider a dynamics-aware fairness intervention based on feedback control that parametrically violates classifier group-independence (and therefore, in our setting, *equalized odds* [12–14] and *envy-freeness* [17, 18]) to achieve long-term fairness. We use simulation to contrast this feedback control policy to a group-independent classifier; a policy subject to demographic parity [10, 11]; and "laissez-faire", *group-specific* policies. We conclude by discussing the limitations of our model and our findings and by outlining potential future work.

## 1.2 Related work

Our work chiefly contributes to the literature on fairness in machine learning but also builds on prior work on "statistical discrimination". The most relevant publications are those that have highlighted

the importance of studying the dynamics and long-term consequences of machine learning, fairness constraints, and models of population response. In particular, Liu et al. [30] use Markov transitions to model agent responses to classification without considering classifier retraining; D'Amour et al. [22] and Zhang et al. [23] reapply Markov transitions to agent attributes in the presense of classifier retraining; Zhang et al. [32] model agents' decisions of whether to engage with classification based on perceived accuracy and intra-group disparity; Coate and Loury [21], Hu and Chen [27], and Liu et al. [26] considered economical "best-response" models to agent labels with classifier updates; and Heidari et al. [24] considered an imitation-based model of social learning in which agents choose between the strategies of other agents to maximize utility and minimize effort. Tang et al. [33] also studied the delayed and accumulated impacts of past deployed policies, but did not study the fairness implication of such impacts. Similarly, literature on "fair bandit/reinforcement learning" [34–36] has largely focused on technical aspects of imposing normative present fairness in a sequential setting.

Our proposed model synthesizes prior conceptual innovations: First, Coate and Loury [21], Hu and Chen [27], and Ensign et al. [37] each considered incomplete information available to a classifier as a means to equitably endogenize persistent predictor bias, but did not consider incomplete information available to individual agents. Second, the class of response functions considered by Mouzannar et al. [28] allows group-level strategic responses to depend on existing qualification rates and may be used to endogenize persistent disparity under group-independent policies; the cited work does not explore this direction, but, like us, the authors assume "groups are ex-ante equal in all respect except for their qualification profiles...and any potential coupling between groups can only happen through the different and interacting selection rates induced by the policies" [28, p. 362]. Atop this foundation, we provide a plausible mechanism of imitation, motivated by incomplete information available to individual agents, to justify replicator dynamics as a special case of such response functions, and we extend a dynamical analysis for a classifier forced to contend with misclassification errors.

To support our use of the replicator equation to model group-level responses to classification, we cite the imitation-based derivation(s) of the replicator equation by Björnerstedt and Weibull [38]; the characterization of evolutionarily stable strategies conducted by Taylor and Jonker [39]; the analogy of memes as attributed to Dawkins [40]; and the extensive application of the replicator equation in game-theoretic contexts as explored by Friedman and Sinervo [41].

## 2   Formulation

*We defer all proofs and provide them in Appendix B of the supplementary material.*

We consider countably many *agents*, $n \geq 2$ *groups*, and a single *classifier*. Until Section 2.2, our setting matches that of Coate and Loury [21] but treats $n$ groups and a more granular classifier utility function. We ground statements with a running example: a regional bank (classifier) serving several isolated communities (groups, subpopulations) by offering standardized loans for which every individual (agent) applies. Alternative examples include hiring decisions [21] or college admissions.

Agents belong to **groups**, interpreted in Section 2.2 and consistent with *isolated communities* in our running example, with known relative frequencies $\mu_g \in (0, 1)$. We vectorize these frequencies as $\boldsymbol{\mu}$.

$$\mathcal{G} \coloneqq \{1, 2, ..., n\}; \quad \forall g \in \mathcal{G}, \, \mu_g \coloneqq \Pr(G = g); \quad \sum_{g \in \mathcal{G}} \mu_g = 1; \quad \boldsymbol{\mu} \coloneqq (\mu_1, \mu_2, ..., \mu_n) \quad (1)$$

*For all statements of probability, we assign uniform probability mass to each agent.*

In addition to relative size $\mu_g$, each group has a **qualification rate** $s_g \in (0, 1)$, which we vectorize as our **state** variable $\mathbf{s}$. We denote the global qualification rate as $\overline{s}$:

$$s_g \coloneqq \Pr(Y = 1 \mid G = g); \quad \mathbf{s} \coloneqq (s_1, s_2, ..., s_n); \quad \overline{s} \coloneqq \sum_{g \in \mathcal{G}} \mu_g s_g = \langle \boldsymbol{\mu}, \mathbf{s} \rangle \quad (2)$$

**Assumption 1.** *No community is completely (un)qualified;* $\quad \forall g, \, s_g \in (0, 1)$.

In our banking example, a qualified ($Y = 1$) individual will repay a loan in full if accepted ($\hat{Y} = 1$), and we presume this outcome to be desirable. The fraction of qualified individuals in community $g$ is represented by $s_g$. Assumption 1 states that no community is completely (un)qualified, and, because $\mu_g \in (0, 1)$, neither is the total population, *i.e.*, $\overline{s} \in (0, 1)$.

Table 1: Agent-specific variables forming a Markov chain.

| Variable | Meaning | Domain | Realizations |
|---|---|---|---|
| $G$ | group | $\mathcal{G} = \{1, 2, ..., n\}$ | $g, h, i, j$ |
| $Y$ | qualification | $\{0, 1\}$ *i.e.*, {unqualified, qualified} | $y$ |
| $X$ | feature | $(-\infty, \infty)$ | $x$ |
| $\hat{Y}$ | classification | $\{0, 1\}$ *i.e.*, {reject, accept} | $\hat{y}$ |

The **feature** $X$ of an agent qualified as $(Y = y)$ is sampled according to a probability density function $q_y$. In our banking example, we may interpret $X$ as a "credit score" known to the bank.

$$q_y(x) := p_X(x \mid Y = y); \quad y \in \{0, 1\} \tag{3}$$

**Assumption 2.** *The qualification-conditioned distribution of features $q_y(x)$ is group-independent.*

Assumption 2 ensures that qualified individuals are statistically indistinguishable in terms of feature $X$ across different communities—as are unqualified individuals. Given an agent's qualification $Y$, learning $G$ gives no additional information about $X$.

**Assumption 3.** *$q_y$ is differentiable and strictly positive for each $y$. The values of $X$ are ordered and unified such that $q_1(x)/q_0(x)$ is strictly increasing in $X$:*

$$\forall x, y, \; q_y(x) \in (0, \infty); \quad \frac{d}{dx}\left(\frac{q_1(x)}{q_0(x)}\right) > 0 \tag{4}$$

Assumption 3 ensures that the feature $X$ is "well-behaved": In our example, as credit scores increase as $x$, the odds that individuals with that credit score $x$ will pay off loans also increases.

Finally, a **classifier** observes the feature $X$ of each agent, from which it must predict the agent's correct label $Y$ using a deterministic policy $\pi$ that, unless otherwise stated, remains ignorant of $G$.

**Assumption 4.** *The classifier learns the true distribution $\Pr(Y \mid X)$ before choosing policy $\pi$.*[1]

**Assumption 5.** *The classifier maximizes its expected utility $u$ with risk-neutral preferences. This utility $u$ is linear in each outcome fraction $\Pr(Y = y, \hat{Y} = \hat{y})$, and the coefficients[2] $V_{y,\hat{y}} \in (-\infty, \infty)$ are independent of feature value $X$ and group membership $G$. The classifier receives higher utility from correct predictions ($\hat{Y} = Y$).*

$$\hat{Y} := \pi(X); \quad u(\pi) := \sum_{y,\hat{y}=0}^{1} V_{y,\hat{y}} \Pr(Y = y, \pi(X) = \hat{y}); \quad V_{y=\hat{y}} > V_{y\neq\hat{y}} \tag{5}$$

In our example, Assumption 5 is consistent with a bank maximizing expected net profit, where the bank expects net profit proportional to $V_{y,\hat{y}}$ from each individual qualified as $y$ and approved as $\hat{y}$, independent of credit score $X$ or community $G$. By Assumption 4, the bank selects policy $\pi$ knowing the stochastic relationship between qualification $Y$ and credit score $X$ for the region it serves.

With group-independent classifier policies, having excised assumptions of inherent differences between groups in our formulation, we emphasize that unequal group qualification rates cause any statistical group-level disparities of prediction outcomes. We therefore consider eliminating differences in group qualification rates as a realization of long-term fairness in this setting.

**Theorem 1.** *Discounting sets of measure zero, the $u$-maximizing, group-independent policy $\pi$ is parameterized by the **feature threshold** $\phi \in [-\infty, \infty]$ such that $\pi(x) = 1$ if and only if $x > \phi$, where $\phi$ depends only on the global qualification rate $\bar{s}$.*

$$\hat{y} = \pi(x) = \begin{cases} 1 & x > \phi \\ 0 & otherwise \end{cases}; \quad \frac{q_1(\phi)}{q_0(\phi)} = \xi \cdot \frac{1 - \bar{s}}{\bar{s}}; \quad \xi := \frac{V_{0\hat{0}} - V_{0\hat{1}}}{V_{1\hat{1}} - V_{1\hat{0}}} \tag{6}$$

*When a solution in $\phi$ to the **threshold equation**, Eq. (6), does not exist, $\phi$ is either $\pm\infty$.*

**Corollary 1.1.** *The classifier's feature threshold $\phi$ responds inversely to $\bar{s}$: $\frac{d\phi}{d\bar{s}} < 0, \quad \frac{d\bar{s}}{d\phi} < 0$.*

---

[1] In practice, this distribution may be learned from sufficient data.

[2] We will abuse notation to write, *e.g.*, $V_{y,\hat{y}}$ as $V_{1\hat{0}}$, to disambiguate the order of indices on $V$ and, later, $U$.

**Assumption 6.** *V is such that $\xi \in (0, \infty)$ ($\xi$ is defined in Theorem 1, Eq. (6)).*

The threshold equation, Eq. (6), is a reprise of Coate and Loury [21] restricted to group-independent classifier policies. In our example, the bank maximizes its utility by approving individuals with credit scores greater than $\phi$ and denying everyone else. Interpreting Assumption 6, there exist populations for which the bank prefers to accept some applicants and reject others.

## 2.1 Time-dependence

We model our system in discrete time for semantic reasons, acknowledging that a learning process consistent with Assumption 4 requires time, although the mathematics generalize to continuous time without issue.[3] Where required, we will denote time-dependence in square brackets $[t]$. Where we omit this explicit dependence, as in all prior expressions, it is understood that all variables in an expression correspond to the same time $t$.

**Assumption 7.** *The relative sizes of groups $\mu_g$, qualification-conditioned feature distributions $q_y$, and classifier utility coefficients $V_{y,\hat{y}}$ are all time-independent. All prior assumptions hold independently for each time step.*

## 2.2 Replicator dynamics

Anticipating algorithmic classification, how do agents decide whether to become qualified? In our banking example, we imagine that individuals must "invest" or "apply capital" to be able to pay back loans and that the "rationality" of doing so depends on what they know about the classification policy and potential outcomes—which they must estimate from incomplete information provided by peer examples. To deal with the uncertainty of limited examples, we imagine the emergent heuristic of updating personal qualification by *imitating* the strategies of others based on popularity and "success". For example, if your friend chose to become qualified for a loan and now runs a small business, the success of the business may induce you to seek qualification yourself by first building credit history; if many of your neighbors receive loans despite being unqualified and appear successful investing in speculative assets, you may infer that qualification is a waste of resources.

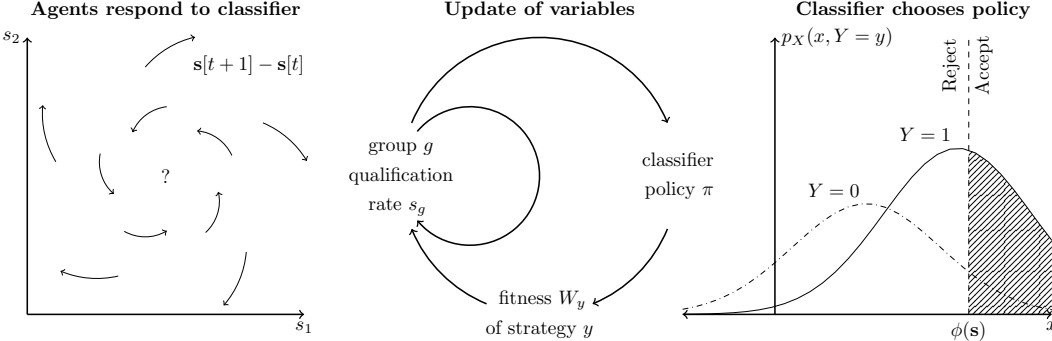

Figure 1: Our model appeals to the replicator equation Eq. (7) to model population response and considers a Bayes-optimal, group-independent classifier policy $\pi$ with feature threshold $\phi$ (Eq. (6), right pane). When coupled (middle pane), these equations give rise to an autonomous dynamical system. We wish to understand how the vector of group qualification rates **s**, as our state variable, changes in time (left pane).

Björnerstedt and Weibull [38] have shown that *imitation* in this form, whereby agents stochastically update to strategies weighted by success and popularity[4] yields the (continuous time) **replicator equation**, which we use in its discrete time form, as detailed by Friedman and Sinervo [41]:

$$s_g[t+1] = s_g[t]\frac{W_1[t]}{\overline{W}_g[t]}; \quad \overline{W}_g := W_1 s_g + W_0(1 - s_g); \quad \forall y, \; W_y \geq 0 \qquad (7)$$

---

[3]The continuous replicator equation appears in Björnerstedt and Weibull [38] & Friedman and Sinervo [41].

[4]Weighting by popularity effects "preferential attachment" for the success of a strategy in a subpopulation and thus introduces dynamical inertia. Replicator dynamics also arise when agents update strategies with (Poisson distributed) expected periodicity that is affine in the success of one's current strategy [38].

Here, $W_y$ is the **fitness** of strategy $y$, which, by Theorem 2, we model as independent of feature $X$ and group $G$. Following Björnerstedt and Weibull [38], we may derive the fitness $W_y$ in terms of expected "success" $U_{y,\hat{y}}$ of each qualification-classification outcome $(y, \hat{y})$:

**Assumption 8.** *The fitness of strategy $(Y = y)$, denoted as $W_y^g$, is affine in the average success $U_{y,\hat{y}}$ of qualification $(Y = y)$ with classification $(\hat{Y} = \hat{y})$. $U_{y,\hat{y}}$ is time-, feature- and group-independent. Without loss of generality, we restrict $U_{y,\hat{y}} \in [0, \infty)$ and drop the constant bias term from each $W_y^g$.*

$$W_y^g := \sum_{\hat{y}=0}^{1} \Pr(\hat{Y} = \hat{y} \mid Y = y, G = g) U_{y,\hat{y}} \tag{8}$$

**Theorem 2.** *The fitness $W_y^g$ of strategy $Y = y$ in group $g$ is feature- and group-independent.*

$$\forall y, g, \ W_y^g = W_y \tag{9}$$

Intuitively, "success" $U$ may be interpreted as *utility* or *payoff* to each agent when agents align strategy adoption with personal incentives, but, fundamentally, $W$ corresponds to the relative success of the *strategy* in replicating, *i.e.*, spreading between individuals. The strategies of (non)qualification are thus subject to evolutionary pressures, competing to out-replicate each other in an environment shaped by perceptions of classifier policy. Notably, the fitness of a strategy depends only on the classifier—not group membership $G$. Agents remain identically modelled across all groups.

**Assumption 9.** *The success of (non)qualification is sensitive to classification, and the expected success for qualified individuals increases with classifier acceptance: $U_{0\hat{1}} \neq U_{0\hat{0}}$; $U_{1\hat{1}} > U_{1\hat{0}}$ .*

**Assumption 10.** *Each group $g$ has the properties of a closed population in which* qualification*, as a strategy or meme [40], competes with* non-qualification *free from exchange with other groups.*

It is significant that we model population updates as independent for *closed* populations, as this restricts our interpretation of groups, which must be functionally impermeable to the exchange of qualification strategies. To precisely delineate between real-world examples of "groups" is akin to disassociating "cultures", which also imply boundaries of exchange but generally intersect. Having noted that "sensitive attributes" such as race, sex, color, *etc.* may not correspond to meaningful divisions between people (which depend on social context), we instead qualify a **group** by the extent to which it satisfies Assumption 10. As an open question, we ask whether imposing arbitrary demographic-dependent policies may catalyze the formation of groups of strategic peers, but we will consider insular social groups or isolated communities as canonical examples.

## 3 Dynamics

The threshold equation, Eq. (6): $\phi[t](\mathbf{s}[t])$, and the replicator equation, Eq. (7): $\mathbf{s}[t + 1](\phi[t], \mathbf{s}[t])$, may be coupled to yield an autonomous dynamical system $\mathbf{s}[t + 1](\mathbf{s}[t])$ that evolves in time. To analyze it, we first generate a useful set of coordinates to compliment $\bar{s}$ and track qualification rate disparities, defined by the differences in $s_g$ between groups. We then note the importance of $W_1(\phi) - W_0(\phi)$ to the overall dynamics of the system, and use it to identify non-trivial equilibrium states. To interpret this section for our example, we ask how community-specific loan qualification rates change as individuals imitate successful strategies in their isolated communities, while assuming that the bank maximizes profit using group-independent credit thresholds for loan approval.

**Definition 3.** Define the (signed) **qualification distance** from group $h$ to group $g$ as

$$\delta(g, h) := s_g - s_h, \quad g, h \in \{1, 2, ..., n\} \tag{10}$$

We next define the vector $D$ comprising $(n - 1)$ linearly-independent qualification distances between sequential pairs of subpopulations:

$$D := \Big( \delta(1, 2), \ \delta(2, 3), \ ..., \ \delta(n - 1, n) \Big) \tag{11}$$

The components of $D$ and value of $\bar{s}$ together yield a complete set of coordinates to describe the state of the dynamical system, which we may exchange for the original vector of qualification rates $\mathbf{s} = (s_1, s_2, ..., s_n)$ via a non-orthogonal, linear change of basis (See Appendix B):

$$s_g = \bar{s} + \sum_{h=g}^{n-1} \delta(h, h + 1) - \sum_{h=1}^{n-1} \sum_{k=1}^{h} \mu_k \delta(h, h + 1) \quad \forall g \in \mathcal{G} \tag{12}$$

Let us denote the state vector in our new coordinate system as $\mathbf{r} := \big( \delta(1, 2), \delta(2, 3), ..., \delta(n-1, n), \bar{s} \big)$.

**Definition 3.1.** For $p \geq 1$, define a state's **$p$-total qualification rate disparity** as the $p$-norm of $D$:

$$\|D\|_p := \Big( \sum_{g=1}^{n-1} |\delta(g, g+1)|^p \Big)^{1/p} \tag{13}$$

**Remark 4.** States $\mathbf{s}$ with a common $\bar{s}$ value form a **hyperplane** $\bar{s} = \langle \boldsymbol{\mu}, \mathbf{s} \rangle$ (Eq. (2)), by definition.

**Theorem 5.** *The nullity of any $p$-total qualification rate disparity is preserved in time.*

$$p \geq 1; \quad \|D[t]\|_p = 0 \iff \|D[t+1]\|_p = 0 \tag{14}$$

Theorem 5 highlights a weak notion of the persistence of disparity within the system sans intervention: Any state that possesses some non-zero total qualification disparity (defined as some chosen $p$-norm of $D$) must always exhibit some non-zero total qualification disparity with any finite time horizon. In our example, if some communities start more qualified than others, the qualification rates of different communities will not naturally equalize in any given lifetime. Note that this statement is insufficient to address the limit $t \to \infty$, however. For a stronger result that includes this limit (Theorem 10), we first characterize the system's equilibrium states.

## 3.1 Equilibrium

**Definition 6.** The system as a whole is **at equilibrium** when, for all $g \in \mathcal{G}$ simultaneously, $s_g$ is stationary in time:

$$\text{at equilibrium} \xleftrightarrow{\text{def}} \forall g \in \mathcal{G}, \ \exists t_0 \text{ s.t. } \forall t \geq t_0, \quad s_g[t] = s_g[t_0] \tag{15}$$

*Note*: replicator dynamics is an instance of the more general family of monotone dynamics, with which all equilibria are shared [38, 41].

**Theorem 7.** *Disregarding boundary states by Assumption 1, the replicator equation, Eq. (7), implies*

$$\text{sgn}\big(\bar{s}[t+1] - \bar{s}[t]\big) = \text{sgn}\big(W_1(\phi[t]) - W_0(\phi[t])\big) \tag{16}$$

**Theorem 8.** *It is necessary and sufficient for a system at equilibrium that $W_1 = W_0$ or for the system to occupy some vertex of the state space.*

$$\text{at equilibrium} \iff \begin{cases} W_1 = W_0 & \textit{(internal equilibrium)} \\ \forall g \in \mathcal{G}, \quad s_g \in \{0, 1\} & \textit{(trivial equilibrium)} \end{cases} \tag{17}$$

Theorem 8 indicates that the conditions for internal equilibrium are described by the *zeros* of the function $W_1(\phi) - W_0(\phi)$, as depicted in Fig. 2, and, by the threshold equation, Eq. (6), $\phi$ has dynamical dependence only on $\bar{s}$. It follows that only certain values of $\bar{s}$ support internal equilibrium, and each value corresponds to a hyperplane in state space (Remark 4).

**Theorem 9.** $W_1(\phi) - W_0(\phi)$ *is strictly quasi-concave in $\phi$. This guarantees that no more than two zeros of the function $W_1 - W_0$ exist.*

We denote the possible zeros of $W_1 - W_0$ as $\phi^+$ and $\phi^-$, where the sign in the superscript indicates the local slope of the function. These zeroes correspond to parallel hyperplanes in state space that comprise all interior equilibria

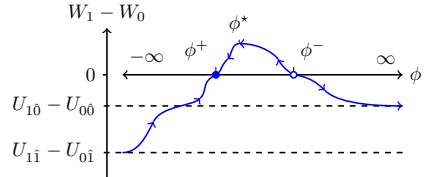

Figure 2: $W_1(\phi) - W_0(\phi)$ (blue curve) is a strictly quasi-concave function of $\phi$. $\phi^\star$ denotes the unique local extremum. The direction of the arrows is a consequence of Theorem 7 and Corollary 1.1.

of the system. Whether $\phi^\pm$ corresponds to an (un)stable equilibrium hyperplane may be determined by Theorem 7 and the sign of $\frac{\partial}{\partial \phi}(W_1 - W_0)$: Only $\bar{s}(\phi^+)$ is stable, and we will verify this fact with linear stability analysis.

**Theorem 10.** *If the state of the system asymptotically approaches an internal equilibrium, the nullity of $p$-total qualification rate disparity is preserved in the limit of infinite time.*

$$p \geq 1; \quad \lim_{t' \to \infty} (W_1 - W_0) = 0 \implies \Big( \|D[t]\|_p = 0 \iff \lim_{t' \to \infty} \|D[t']\|_p = 0 \Big) \tag{18}$$

Theorem 10 formalizes the critical observation that any state that attracts to the stable equilibrium hyperplane, unless initially free from qualification disparity, will forever exhibit some total qualification disparity. This is a more robust notion of the **persistence of disparity** in our system than Theorem 5.

## 3.2 Stability

For our regional banking example, we may imagine that qualification rates settle into a stable pattern in which some communities have a higher average qualification rate than others. How robust is this pattern of inequality to small fluctuations of qualification rates? Using **linear stability analysis** (*i.e.*, linearizing the response of the system to small perturbations about equilibrium and asking "do perturbations amplify or dissipate?"), we show that only the $\phi^+$-hyperplane acts as a stable attractor.

First, let us denote the evaluation of an expression *at equilibrium* by placing a vertical line to the right of the expression with "eq" as a subscript. In light of Theorem 8 and Eq. (7), we also introduce the shorthand $W_{\mathrm{eq}}$ to denote an equilibrium value of $W_1$, $W_0$, or, equivalently, any $\overline{W}_g$. It should be noted that the value of $W_{\mathrm{eq}} \in [0, \infty)$ still depends on the particular equilibrium state of the system.

$$W_{\mathrm{eq}} := W_0 \Big|_{\mathrm{eq}} = W_1 \Big|_{\mathrm{eq}} = \overline{W}_g \Big|_{\mathrm{eq}} \quad \forall g \in \mathcal{G} \tag{19}$$

We linearize the system at equilibrium by constructing the Jacobian $J \in \mathbf{R}^{n \times n}$ corresponding to discrete time-evolution and identifying its eigenvectors and eigenvalues:

$$J := \left[ \frac{\partial \mathbf{r}}{\partial \delta(1,2)} \; \frac{\partial \mathbf{r}}{\partial \delta(2,3)} \; \cdots \; \frac{\partial \mathbf{r}}{\partial \delta(n-1,n)} \; \frac{\partial \mathbf{r}}{\partial \overline{s}} \right] \tag{20}$$

where $\mathbf{r}$, the state vector in $(D, \overline{s})$ coordinates, is interpreted as a column vector.

**Theorem 11.** *The Jacobian $J$ simplifies to a scalar multiplied by a matrix with a single non-zero column $\mathbf{v}$ in the last position.*

$$J \Big|_{\mathrm{eq}} = \frac{1}{W_{\mathrm{eq}}} \left( \frac{d\phi}{d\overline{s}} \right) \left( \frac{d}{d\phi}(W_1 - W_0) \right) \left[ \mathbf{0}^{(n \times n-1)} \Big| \mathbf{v} \right], \quad \mathbf{v} := \begin{bmatrix} \delta(1,2)(1-s_1-s_2) \\ \delta(2,3)(1-s_2-s_3) \\ \cdots \\ \delta(n-1,n)(1-s_{n-1}-s_n) \\ \sum_{g \in \mathcal{G}} \mu_g s_g (1-s_g) \end{bmatrix} \tag{21}$$

The eigenvalues of $J$ determine the stability of the system at equilibrium.

**Corollary 11.1.** *At equilibrium, any state displacement vector with zero $\overline{s}$ component is an eigenvector of $J$ with eigenvalue 0, while $\mathbf{v}$ is an eigenvector of $J$ with eigenvalue $\lambda$:*

$$\lambda := \left( \sum_{g \in \mathcal{G}} \mu_g s_g (1-s_g) \right) \frac{1}{W_{\mathrm{eq}}} \left( \frac{d\phi}{d\overline{s}} \right) \left( \frac{d}{d\phi}(W_1 - W_0) \right) \Big|_{\mathrm{eq}} \tag{22}$$

Perturbing (displacing) a state vector $\mathbf{r}$ at an internal equilibrium by altering any combination of coordinates appearing in $D$—while leaving $\overline{s}$ fixed—specifies motion on the $\overline{s}$ hyperplane occurring in neutrally stable equilibrium (*i.e.*, a displacement vector with zero $\overline{s}$ component has null eigenvalues at internal equilibrium. See Strogatz [42]). An internal equilibrium is stable to perturbations in $\mathbf{v}$, leaving the hyperplane, iff $\lambda$ is negative (and, in discrete-time, $> -2$ to forbid over-corrections) [42].

**Corollary 11.2.** *As a consequence of Corollary 1.1, which states $\frac{d\phi}{d\overline{s}} < 0$, the eigenvalue $\lambda$ in Eq. (22) is negative, (and the associated equilibrium hyperplane stable) iff $\frac{d}{d\phi}(W_1 - W_0)|_{\mathrm{eq}} > 0$. This prescribes precisely the value $\phi^+$ for the stable equilibrium hyperplane.*

## 4 Interventions

In the dynamical setting we have characterized, we now explore "fairness interventions", which substitute the set of policies that the classifier may choose from, possibly permitting group-specific decision rules $\pi_g$. We first observe that for the default policy with a group-independent feature threshold $\phi$, one commonly cited standard of normative present fairness is automatically satisfied.

**Definition 12. Equalized Odds** [12–14] requires that a classifier's decisions $\hat{Y}$, given by policy $\pi$, misclassify (un)qualified agents at equal rates across groups:

$$\forall g, h \in \mathcal{G}, \; \forall y, \hat{y} \in \{0, 1\}, \quad \Pr(\hat{Y} = \hat{y} \mid Y = y, G = g) = \Pr(\hat{Y} = \hat{y} \mid Y = y, G = h) \tag{23}$$

**Theorem 13.** *For policies defined by group-specific thresholds $\phi_g$, the equivalence of these feature thresholds $(\forall g, \phi_g = \phi)$ is necessary and sufficient to satisfy Equalized Odds given the group-independence of each $q_y$ (Assumption 2).*

By Theorem 13, a group-independent policy satisfies Equalized Odds (*e.g.*, the bank accepts/rejects (un)qualified loan applicants at group-independent rates), yet disparities may persist (Theorem 10). This indicates a counter-example to reliance on Equalized Odds for long-term fairness in our model, *viz.*, the optimal group-independent threshold classifier we have studied so far.

**Corollary 13.1.** Equalized Odds does not imply long-term fairness in our model.

We next ask whether a small displacement a group-independent threshold $\phi$ near the $\phi^+$-hyperplane, which we interpret as a **universal subsidy** (or penalty), can diminish qualification rate disparities.

**Theorem 14.** $\Theta(\epsilon)$ *perturbations of a group-independent $\phi$ at internal equilibrium induce motion, which, to first-order approximation (i.e., ignoring $\mathcal{O}(\epsilon^2)$ terms), is parallel to the eigenvector* **v**.

As a consequence of Theorem 14, while **v** need not be orthogonal to the equilibrium hyperplane, and a universal subsidy may decrease qualification rate disparity while applied (settling on a new equilibrium hyperplane with different, though persistent disparities), the system is stable to such per-turbations at $\phi^+$ as characterized by linear system response and will relax to the original equilibrium state when the intervention is removed. To permanently change qualification disparities, a temporary universal subsidy (penalty) must rely on the **non-linear response** of the system and is therefore liable to require *large* perturbations to the classifier's threshold $\phi$. This finding compels us to consider interventions with group-dependent threshold perturbations—or group-dependent thresholds. To this end, we hereafter generalize our classifier such that it independently classifies each group $g$ according to a group-specific threshold $\phi_g$. We denote the vector of these thresholds as $\Phi := (\phi_1, \phi_2, ..., \phi_n)$ and assume that, prior to some perturbative intervention, $\phi_g = \phi$ for each $g \in \mathcal{G}$.

**Definition 15. Demographic Parity** [10, 11] requires that a classifier's decisions $\hat{Y}$, given by policy $\pi$, are positive ($\hat{Y} = 1$, *e.g.*, accepting a loan application) at equal rates for all groups:

$$\forall g, h \in \mathcal{G}, \quad \Pr(\hat{Y} = 1 \mid G = g) = \Pr(\hat{Y} = 1 \mid G = h) \tag{24}$$

**Definition 16. Laissez-Faire** allows a separate, $u$-maximizing threshold $\phi_g$ for each group.

**Theorem 17.** *Demographic parity requires sign-heterogeneous, group-dependent changes to the Laissez-Faire values of $\phi_g$ when $\pi$ is non-trivial (does not uniformly accept (reject)).*

Satisfying demographic parity in our setting requires the solution of a differential equation in $q_y$ (Appendix B), which we do not rigidly constrain. We therefore rely on numerical simulation, rather than analytical tools, to evaluate this intervention for our system.

**Feedback control** Arbitrary state transitions in the equilibrium hyperplane may be permanently effected by group-dependent perturbations to $\Phi$, which we derive from linear system response at equilibrium. Specifically, to diminish a specific qualification distance $\delta(g, g+1)$ for given $g$, $\Phi$ may be perturbed by a vector quantity $\Delta_g \Phi = (\Delta_g \phi_1, \Delta_g \phi_2, ..., \Delta_g \phi_n)$.

**Theorem 18.** *On the stable internal equilibrium hyperplane, infinitesimal perturbation of $\Phi$ by*

$$\Delta_g \Phi := -\epsilon \delta(g, g+1) \left( \frac{\alpha_g}{s_1(1-s_1)}, ..., \frac{\alpha_g}{s_g(1-s_g)}, \frac{\beta_g}{s_{g+1}(1-s_{g+1})}, ..., \frac{\beta_g}{s_n(1-s_n)} \right) \tag{25a}$$

$$\alpha_g := (\mu_{g+1} + \mu_{g+2} + ... + \mu_n), \qquad \beta_g := -(\mu_1 + \mu_2 + ... + \mu_g) \tag{25b}$$

*will induce motion in the system preserving $\overline{s}$ and each $\delta(h, h+1)$ for $h \neq g$. The value of $\delta(g, g+1)$ will be diminished by a ratio proportional to the **strength parameter** $\epsilon > 0$.*

Perturbations of the form $\Delta_g \Phi$ may be composed linearly for multiple values of $g$. In particular, when $\epsilon$ is a universal quantity, we may determine the total perturbation to $\Phi$ necessary to simultaneously and proportionately decrease all qualification distances for any given state on the stable equilibrium hyperplane. Let us denote this total perturbation as $\Delta\Phi := \sum_{g \in \mathcal{G}} \Delta_g \Phi = (\Delta\phi_1, \Delta\phi_2, ..., \Delta\phi_n)$. Component-wise, $\Delta\Phi$ is given by

$$\Delta\phi_g = \frac{-\epsilon}{s_g(1-s_g)} \left( \sum_{h=g}^{n-1} \alpha_h \delta(h, h+1) + \sum_{h=1}^{g-1} \beta_h \delta(h, h+1) \right) \tag{26}$$

We note that the proposed feedback control mechanism depends only on the known constants $\mu_g$ and feedback in terms of current qualification distances $\delta$. In addition, the force of the intervention can be tuned by setting the strength parameter $\epsilon$. Finally, we remark that this mechanism can be composed with global perturbations of $\phi$, *i.e.*, universal subsidy in the manner of Theorem 14, to intervene without rejecting any agents that would have been accepted under a group-independent policy.

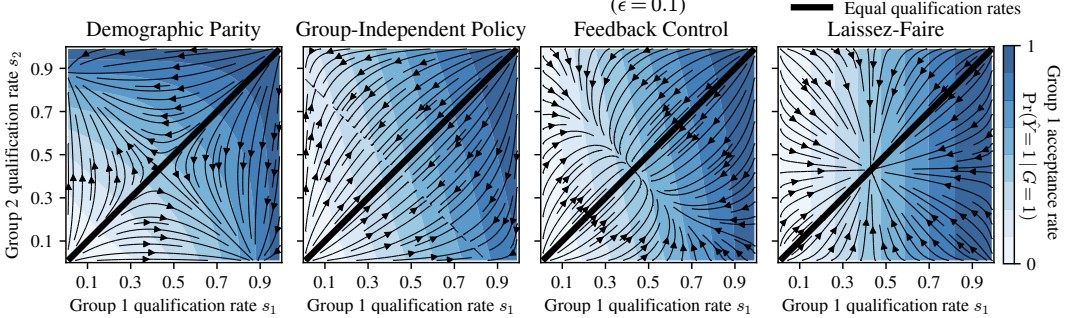

Figure 3: Simulated dynamics for two groups of equal size, subject to different global interventions. Streamlines approximate system time evolution. $q_0$ and $q_1$ are Gaussians with unit variance and have means $-1$ and $1$, respectively. Other examples and rendered dynamical variables are provided in Appendix C. For this example, $(U_{0\hat{0}} = 0.1; U_{0\hat{1}} = 5.5; U_{1\hat{0}} = 0.5; U_{1\hat{1}} = 1.0; V_{0\hat{0}} = 0.5; V_{0\hat{1}} = -0.5; V_{1\hat{0}} = -0.25; V_{1\hat{1}} = 1.0)$.

We compare interventions by appeal to simulation, choosing a setting that guarantees a single, stable average qualification rate $\overline{s}^\star$ under group-independent policies (**GI**) (Fig. 3). We consider trade-offs between *normative present fairness* ($\mathbf{F_{NP}}$) (*e.g.*, demographic parity (**DP**) or equalized odds (**EO**)) and *long-term fairness* ($\mathbf{F_{LT}}$), for which the dynamics must converge to the line demarcating equal qualification rates. Darker shading (blue) implies a higher absolute acceptance rate for Group 1, which, by the setting's symmetry, is the same for Group 2 when reflected across the aforementioned line; reflectional asymmetry violates DP. Under compulsory DP (first pane), the system violates $F_{LT}$, settling into a "patronizing equilibrium under affirmative action", as coined by Coate and Loury [21], in which agents from a less-qualified group are *patronized* (*e.g.*, granted loans despite nonqualification) by the classifier (*cf.* the upper-left corner, with low qualification and high acceptance rates for Group 1). States under GI (second pane), which satisfies the EO notion of $F_{NP}$ by Theorem 13, converge to a line of constant $\overline{s}$ (*cf.* Remark 4) while preserving qualification disparities (*cf.* Theorem 10). $F_{LT}$ is expected from a **laissez-faire** (**LZ**) policy (last pane), which adopts group-specific policies and thus decouples all group dynamics: Each group must converge to $\overline{s}^\star$ separately. Still, LZ satisfies neither the DP (by reflectional asymmetry) nor EO (by Theorem 13) notions of $F_{NP}$. In contrast, feedback control (third pane) achieves $F_{LT}$ by conceding $\epsilon$-small, parametric violations of $F_{NP}$ (EO) (See Appendix C for plots of classifier error rates in this setting).

## 5  Discussion and limitations

The novelty of our contribution is the demonstration of persistent qualification rate disparities in a symmetric setting consistent with plausible mechanisms of population response—sustained by the careless deployment of machine learning and myopic fairness interventions. We submit that, given the many charitable assumptions of our model to achieve perfect structural equality between groups, any reasonable fairness intervention should succeed in responsibly rectifying disparities here, if anywhere. Moreover, we have laid bare inherent tensions that can exist between the *means* and *ends* of fairness considerations in a dynamical context, demonstrating the potential incompatibility of immediate and long-term notions of fairness.

We acknowledge that our model is simplistic, but such simple cases must be well-understood as a first step towards further, equitable models of population response. We regard the requirement of strictly isolated groups as the most tenuous assumption of our model and conjecture that even relatively weak inter-group exchange of strategies should lead to long-term fairness in our default setting. Nonetheless, we believe that a program based on incomplete agent information can successfully endogenize persistent disparities in symmetric settings more robustly. Specifically, future work may consider multiple classifiers with different task domains affecting a common population; we expect this extension to readily endogenize broken symmetries between group environments and conditions. We also trust that voluntary participation, as considered by Zhang et al. [32], may be modelled as an additional strategy within our framework. Regarding empirical falsifiability, we note that the dynamics of social disparity are not exclusive to algorithmic classifiers [27], and ask whether our model's predictions may be contrasted with existing and historical resource allocation problems.

We invite readers to consider both our model and application of control theory to society through algorithmic classification, with care. We intend our work to *reform* the misapplication of machine learning, inappropriate modelling assumptions, and myopic notions of fairness.

## Acknowledgments and Disclosure of Funding

We believe our paper benefited significantly from NeurIPS 2021 review process. We thank our peer-reviewers for their helpful feedback and constructive criticism of the originally submitted version of this paper. We also thank the NeurIPS Area Chairs, Senior Area Chair, Ethics Chairs, and Program Chairs for their discussion of relevant sociotechnical issues and helpful recommendations in preparation of this published version, in which we believe our contributions are better framed and clarified.

This work is partially supported by the National Science Foundation (NSF) Program on Fairness in Artificial Intelligence in Collaboration with Amazon (FAI) under grant IIS-2040800 and by NSF grant CCF-2023495.

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
