The Appendices are organized according to Table 2.

Table 2: Organization of appendices

| Appendix | Content |
|---|---|
| Appendix A | Summary of notation used in the main paper |
| Appendix B | Proofs of all theorems, remarks, and corollaries as well as additional lemmas |
| Appendix C | Figures like Fig. 3, exploring different settings and variables of interest |

# A  Notation

Table 3: Choice of notation

**Parameters:**

| | |
|---|---|
| $n$ | Number of groups |
| $\mathcal{G}$ | Set of groups $\{1, 2, ..., n\}$ |
| $\mu_g$ | Fraction of total population in group $g$ |
| $\boldsymbol{\mu}$ | Vector $(\mu_1, \mu_2, ..., \mu_n)$ |
| $V$ | $2 \times 2$ matrix of classifier utilities, indexed by $(Y, \hat{Y})$ pairs |
| $\theta$ | Classifier's probability threshold (Lemma 1.1) |
| $U$ | $2 \times 2$ matrix of agent fitnesses, indexed by $(Y, \hat{Y})$ pairs |
| $q_y$ | Probability density function of $X$ given $Y = y$ |
| $Q_y$ | Cumulative distribution function of $X$ given $Y = y$ |

**Random Variables:**

| | |
|---|---|
| $G$ | Group to which an agent belongs |
| $X$ | Real-valued feature of an agent |
| $Y$ | Actual binary label (qualification) of an agent |
| $\hat{Y}$ | Predicted binary label (qualification) of an agent |

**Indices:**

| | |
|---|---|
| $g, h, i, j$ | used to indicate a group |
| $x$ | used to indicate a feature value |
| $y$ | used to indicate a binary label (qualification) |
| $\hat{y}$ | used to indicate a predicted binary label (qualification) |

**Dynamical Variables:**

| | |
|---|---|
| $t$ | Discrete time |
| $\cdot[t]$ | Restriction of a dynamical variable to time $t$ |
| $s_g$ | Fraction of qualified agents in group $g$ |
| $\mathbf{s}$ | Vector $(s_1, s_2, ..., s_n)$ |
| $\bar{s}$ | Fraction of qualified agents in total population |
| $\delta(g, h)$ | Difference in qualification rates between groups: $s_g - s_h$ |
| $D$ | A set of $n - 1$ linearly independent qualification distances $\delta$ |
| $\mathbf{r}$ | Vector with the elements of $D$ prepending $\bar{s}$ as components |
| $\pi$ | Classifier's policy mapping $X$ to $\hat{Y}$ |
| $\phi$ | Classifier's feature threshold (Theorem 1) |
| $W_y$ | Average agent fitness conditioned on $Y = y$ |
| $\overline{W}_g$ | Average agent fitness conditioned on $G = g$ |

**Miscellaneous:**

| | |
|---|---|
| $J$ | Jacobian matrix for dynamical system. |
| $\lambda$ | A specific eigenvalue of $J$ |
| $\mathbf{v}$ | A specific eigenvector of $J$ |
| $\Phi$ | Vector of group-specific feature thresholds $\phi_g$ |

# B  Proofs

## Proof of Theorem 1

**Lemma 1.1.** *Discounting sets of measure zero, the $u$-maximizing policy $\pi$ is parameterized by a single **probability threshold** $\theta \in [0, 1]$ such that*

$$\pi(x) = \begin{cases} 1 & \Pr(Y = 1 \mid X = x) > \theta \\ 0 & \text{otherwise} \end{cases} \tag{27}$$

*where*

$$\theta = \frac{V_{0\hat{0}} - V_{0\hat{\imath}}}{V_{1\hat{\imath}} - V_{0\hat{\imath}} + V_{0\hat{0}} - V_{0\hat{\imath}}} \tag{28}$$

*Proof of Lemma 1.1.* Discounting sets of measure zero (*i.e.*, rejecting the possibility of strict equality as infinitely unlikely), a classifier will accept an agent with feature value $X = x$ if and only if the expected utility of doing so is greater than rejecting.

$$\left( \sum_{y=0}^{1} \Pr(Y = y \mid X = x) V_{y\hat{\imath}} \right) > \left( \sum_{y=0}^{1} \Pr(Y = y \mid X = x) V_{y\hat{0}} \right) \tag{29}$$

This reduces algebraically to

$$\frac{\Pr(Y = 1 \mid X = x)}{\Pr(Y = 0 \mid X = x)} > \xi \tag{30}$$

where

$$\xi := \frac{V_{0\hat{0}} - V_{0\hat{\imath}}}{V_{1\hat{\imath}} - V_{1\hat{0}}} \tag{31}$$

and by change of variables

$$\theta := \frac{\xi}{1 + \xi} = \frac{V_{0\hat{0}} - V_{0\hat{\imath}}}{V_{1\hat{\imath}} - V_{1\hat{0}} + V_{0\hat{0}} - V_{0\hat{\imath}}}, \quad \xi = \frac{\theta}{1 - \theta} \tag{32}$$

to

$$\Pr(Y = 1 \mid X = x) > \theta \tag{33}$$

Eq. (33) is thus the sole criterion for accepting an agent with feature value $X = x$, and our proof is complete.

**Lemma 1.2.**

$$\Pr(Y = 1 \mid X = x) = \frac{\bar{s} q_1(x)}{\bar{s} q_1(x) + (1 - \bar{s}) q_0(x)} \tag{34}$$

*Proof of Lemma 1.2.* By Bayes's Theorem,

$$\Pr(G = g, Y = 1 \mid X = x) = \frac{p_X(x \mid G = g, Y = 1) \Pr(Y = 1 \mid G = g) \Pr(G = g)}{p_X(x)} \tag{35a}$$

$$= \frac{q_1(x) s_g \mu_g}{\sum_{h \in \mathcal{G}} \left( s_h q_1(x) + (1 - s_h) q_0(x) \right) \mu_h} \tag{35b}$$

By marginalizing over groups $g$, it follows that

$$\Pr(Y = 1 \mid X = x) = \frac{\sum_{g \in \mathcal{G}} s_g q_1(x) \mu_g}{\sum_{h \in \mathcal{G}} \left( s_h q_1(x) + (1 - s_h) q_0(x) \right) \mu_h} \tag{36}$$

This expression may be simplified to the target statement by substituting from Eq. (1):

$$\bar{s} := \sum_{g \in \mathcal{G}} \mu_g s_g \tag{37}$$

$$(1 - \overline{s}) = \sum_g \mu_g - \sum_{g \in \mathcal{G}} \mu_g s_g \tag{38a}$$

$$= \sum_{g \in \mathcal{G}} \mu_g (1 - s_g) \tag{38b}$$

**Lemma 1.3.** $\Pr(Y = 1 \mid X = x)$ *and* $\Pr(Y = 0 \mid X = x)$ *have support for all* $x$.

$$\forall x \in (-\infty, \infty),\ 0 < \Pr(Y = 1 \mid X = x) < 1 \tag{39}$$

*Proof of Lemma 1.3.* By Assumption 3 and Assumption 1, $\overline{s}q_1(x)$ and $(1 - \overline{s})q_0(x)$ must both be strictly positive. As an immediate consequence of Lemma 1.2 and the identity $\Pr(Y = 1 \mid X = x) = 1 - \Pr(Y = 0 \mid X = x)$, we conclude that both $\Pr(Y = 1 \mid X = x)$ and $\Pr(Y = 0 \mid X = x)$ are greater than zero.

**Lemma 1.4.** $\Pr(Y = 1 \mid X = x)$ *is monotonically increasing in* $X$.

*Proof of Lemma 1.4.* By Assumption 3, we have that

$$\forall y, x, \quad q_y(x) \in (0, \infty) \text{ and } \frac{d}{dx}\left(\frac{q_1(x)}{q_0(x)}\right) > 0$$

While from Lemma 1.2,

$$\Pr(Y = 1 \mid X = x) = \frac{\overline{s}q_1(x)}{\overline{s}q_1(x) + (1 - \overline{s})q_0(x)}$$

By the differentiability and strict positivity of each $q$ (Assumption 3) as well as the strict psitivity of $\Pr(Y = 1 \mid X = x)$ by Lemma 1.3, it is sufficient to show strict positivity of the first derivative of $\Pr(Y = 1 \mid X = x)$ to prove monotonicity. We therefore wish to show

$$\frac{d}{dx}\left(\frac{q_1(x)}{q_0(x)}\right) > 0 \implies \frac{d}{dx}\left(\frac{\overline{s}q_1(x)}{\overline{s}q_1(x) + (1 - \overline{s})q_0(x)}\right) > 0 \tag{40}$$

First, let us define

$$a(x) := \overline{s}q_1(x); \quad b(x) := (1 - \overline{s})q_0(x) \tag{41}$$

Our objective may therefore be rewritten as

$$\left(\frac{(1 - \overline{s})}{\overline{s}}\right)\frac{d}{dx}\left(\frac{a(x)}{b(x)}\right) > 0 \implies \frac{d}{dx}\left(\frac{a(x)}{a(x) + b(x)}\right) > 0 \tag{42}$$

Performing explicit differentiation,

$$\frac{d}{dx}\left(\frac{a(x)}{b(x)}\right) = \left(\frac{a(x) + b(x)}{b(x)}\right)^2 \frac{d}{dx}\left(\frac{a(x)}{a(x) + b(x)}\right) \tag{43}$$

we again rewrite our objective as

$$\frac{(1 - \overline{s})}{\overline{s}}\left(\frac{a(x) + b(x)}{b(x)}\right)^2 \frac{d}{dx}\left(\frac{a(x)}{a(x) + b(x)}\right) > 0 \implies \frac{d}{dx}\left(\frac{a(x)}{a(x) + b(x)}\right) > 0 \tag{44}$$

since $\frac{1 - \overline{s}}{\overline{s}} > 0$ by Assumption 1, this is a necessarily true statement.

**Theorem 1 Statement**. Discounting sets of measure zero, the $u$-maximizing, group-independent policy $\pi$ is parameterized by the feature threshold $\phi \in [-\infty, \infty]$ such that $\pi(x) = 1$ if and only if $x > \phi$, where $\phi$ depends only on the global qualification rate $\overline{s}$ as

$$\xi := \frac{V_{0\hat{0}} - V_{0\hat{1}}}{V_{1\hat{1}} - V_{1\hat{0}}}; \quad \frac{q_1(\phi)}{q_0(\phi)} = \xi\left(\frac{1 - \overline{s}}{\overline{s}}\right) \tag{45}$$

When a solution in $\phi$ to the threshold equation, Eq. (45), does not exist, $\phi$ is either $\pm\infty$.

*Proof of Theorem 1.* By Lemma 1.1, a threshold value for $\Pr(Y = 1 \mid X = x)$ (*i.e.*, $\theta$) is sufficient to characterize the optimal classifier policy $\pi$:

$$\pi(x) = \begin{cases} 1 & \Pr(Y = 1 \mid X = x) > \theta \\ 0 & \text{otherwise} \end{cases} \tag{46}$$

Lemma 1.4 concludes that $\Pr(Y = 1 \mid X = x)$ is monotonic in $X$, and so it follows that whenever $\Pr(Y = 1 \mid X = x)$ achieves the threshold value of $\theta$, it does so at a corresponding, unique feature-threshold value $X = \phi$

$$\forall x, \ \Pr(Y = 1 \mid X = x) = \theta \implies x = \phi \tag{47}$$

such that the classifier's policy is given by

$$\pi(x) = \begin{cases} 1 & x > \phi \\ 0 & \text{otherwise} \end{cases} \tag{48}$$

When such equality between $\Pr(Y = 1 \mid X = x)$ and $\theta$ never occurs, we are free to define

$$\phi = \begin{cases} -\infty & \min_x \left\{ \Pr(Y = 1 \mid X = x) \right\} > \theta \\ \infty & \max_x \left\{ \Pr(Y = 1 \mid X = x) \right\} < \theta \end{cases} \tag{49}$$

so that Eq. (48) remains valid in all cases.
Finally, when equality between $\Pr(Y = 1 \mid X = x)$ and $\theta$ does occur, we may solve for finite $\phi$ by re-expressing $\Pr(Y = 1 \mid X = x)$ according to Lemma 1.2:

$$\theta = \frac{\overline{s} q_1(\phi)}{\overline{s} q_1(\phi) + (1 - \overline{s}) q_0(\phi)} \tag{50}$$

Algebraic manipulations are sufficient to derive Eq. (45), where we appeal to Assumption 1 ($s_g \in (0, 1)$) and Assumption 6 ($\xi \in (0, \infty)$, thus $\theta \in (0, 1)$ by Eq. (32)) to ensure that we do not divide by 0.

### Proof of Corollary 1.1

**Corollary 1.1 Statement**. The classifier's feature threshold $\phi$ responds inversely to $\overline{s}$:

$$\frac{d\phi}{d\overline{s}} < 0, \quad \frac{d\overline{s}}{d\phi} < 0 \tag{51}$$

*Proof of Corollary 1.1.* Let us differentiate Eq. (45) with respect to $\overline{s}$. By the chain rule,

$$\frac{d}{d\phi} \left( \frac{q_1(\phi)}{q_0(\phi)} \right) \left( \frac{d\phi}{d\overline{s}} \right) = \left( \frac{\theta}{1 - \theta} \right) \frac{d}{d\overline{s}} \left( \frac{1 - \overline{s}}{\overline{s}} \right) \tag{52a}$$

$$= \left( \frac{\theta}{1 - \theta} \right) \left( \frac{-1}{(\overline{s})^2} \right) \tag{52b}$$

By Assumption 3, Assumption 6, we observe

$$\frac{d}{d\phi} \left( \frac{q_1(\phi)}{q_0(\phi)} \right) > 0; \quad \left( \frac{\theta}{1 - \theta} \right) > 0; \quad \left( \frac{-1}{(\overline{s})^2} \right) < 0 \tag{53}$$

Therefore, by accounting for the sign of each factor in Eq. (52b) and the relationship between derivatives of inverse functions, we conclude that

$$\frac{d\phi}{d\overline{s}} < 0, \quad \frac{d\overline{s}}{d\phi} < 0 \tag{54}$$

### Proof of Theorem 2

**Definition 2.1.** Define the **cumulative distribution functions** $Q_y$ such that

$$Q_y(\phi) := \int_{-\infty}^{\phi} q_y(x)\,dx, \quad y \in \{0,1\} \tag{55}$$

**Lemma 2.1.** $Q_y(\phi)$ *is group-independent.*

$$Q_y(\phi) = \Pr(\hat{Y} = 0 \mid Y = y) \tag{56}$$

*Proof of Lemma 2.1.*

$$Q_i(\phi) := \int_{-\infty}^{\phi} q_i(x)\,dx \tag{57a}$$

$$= \int_{\{x\,:\,\pi(x)=0\}} p_X(x \mid Y = i) \tag{57b}$$

$$= \int_{\{x\,:\,\pi(x)=0\}} p_X(x \mid Y = i, G = g), \quad \forall g \in \mathcal{G} \tag{57c}$$

$$= \Pr(\hat{Y} = 0 \mid Y = i, G = g), \quad \forall g \in \mathcal{G} \tag{57d}$$

$$= \Pr(\hat{Y} = 0 \mid Y = y) \tag{57e}$$

where Eq. (57b) is a consequence of Theorem 1, and Eq. (57c) follows from Assumption 2.

**Theorem 2 Statement.** The fitness $W_y^g$ of strategy $Y = y$ in group $g$ is feature- and group-independent.

$$\forall y, g, \ W_y^g = W_y \tag{58}$$

*Proof of Theorem 2.* By Assumption 8 and Lemma 2.1,

$$W_y^g = U_{y\hat{1}} + (U_{y\hat{0}} - U_{y\hat{1}})Q_y(\phi), \quad y \in \{0,1\} \tag{59}$$

This expression is also group-independent, and we may denote

$$\forall y, g, \ W_y^g = W_y \tag{60}$$

## Verification of Eq. (12)

**Eq. (12) Statement.**

$$s_g = \bar{s} + \sum_{h=g}^{n-1} \delta(h, h+1) - \sum_{h=1}^{n-1}\sum_{k=1}^{h} \mu_k \delta(h, h+1)$$

*Verification of Eq. (12).* We will verify from Eq. (12) directly that

$$\sum_g s_g \mu_g = \bar{s}; \quad s_g - s_{g+1} = \delta(g, g+1) \tag{61}$$

First, let us verify that $\sum_g s_g \mu_g = \bar{s}$, recalling $\sum_{g=1}^{n} \mu_g = 1$. We recognize that thes first and third unexpanded terms in Eq. (12) are unvarying with $g$, while the second term, when summed, negates the third. That is, despite prescribing a different order of summation, precisely the same values are summed in

$$\sum_{g=1}^{n} \mu_g \sum_{h=g}^{n-1} \delta(h, h+1) = \sum_{h=1}^{n-1}\sum_{k=1}^{h} \mu_k \delta(h, h+1) = \sum_{1 \le i \le j < n} \mu_i \delta(j, j+1) \tag{62}$$

and so, as desired,

$$\sum_{g=1}^{n} \mu_g s_g = \bar{s} \tag{63}$$

Next, we rewrite the definitions of $\alpha_g$ and $\beta_g$ appearing in Theorem 18

$$\alpha_g := \sum_{h=g+1}^{n} \mu_h = (\mu_{g+1} + \mu_{g+2} + ... + \mu_n) \tag{64}$$

$$\beta_g := -\sum_{h=1}^{g} \mu_h = -(\mu_1 + \mu_2 + ... + \mu_g) \tag{65}$$

We note that $\alpha_g - \beta_g = 1$ and may rewrite the change of coordinates given in Eq. (12) as

$$s_g = \bar{s} + \sum_{h=g}^{n-1} \delta(h, h+1) - \sum_{h=1}^{n-1} \sum_{k=1}^{h} \mu_k \delta(h, h+1) \tag{66a}$$

$$= \bar{s} + \sum_{h=g}^{n-1} \delta(h, h+1) + \sum_{h=1}^{n-1} \beta_h \delta(h, h+1) \tag{66b}$$

$$= \bar{s} + \sum_{h=g}^{n-1} \alpha_h \delta(h, h+1) + \sum_{h=1}^{g-1} \beta_h \delta(h, h+1) \tag{66c}$$

from which we may verify that

$$s_g - s_{g+1} = \alpha_g \delta(g, g+1) - \beta_g \delta(g, g+1) = \delta(g, g+1) \tag{67}$$

It follows that Eq. (12) inverts the linear change of coordinates prescribed by the definitions of $\bar{s}$ and $\delta(g, g+1)$.

## Proof of Remark 4

**Remark 4 Statement.** States $\mathbf{s}$ with a common $\bar{s}$ value form a **hyperplane** $\bar{s} = \langle \boldsymbol{\mu}, \mathbf{s} \rangle$ (Eq. (2)), by definition.

*Proof of Remark 4.* This follows from the definition of a hyperplane and Eq. (2)

$$\bar{s} := \sum_{g \in \mathcal{G}} \mu_g s_g = \langle \boldsymbol{\mu}, \mathbf{s} \rangle \tag{68}$$

## Proof of Theorem 5

**Theorem 5 Statement.** The nullity of any $p$-total qualification rate disparity is preserved in time.

$$p \geq 1; \quad \left\| D[t] \right\|_p = 0 \iff \left\| D[t+1] \right\|_p = 0 \tag{69}$$

*Proof of Theorem 5.* We first prove the forward direction (for any $p$), $\mathcal{H} = \{1, 2, ..., n-1\}$

$$\sum_{g=1}^{n-1} \|\delta(g, g+1)[t]\|_p = 0 \implies \delta(h, h+1)[t] = 0 \qquad \forall h \in \mathcal{H} \tag{70a}$$

$$\implies s_h[t] = s_{h+1}[t], \quad \overline{W}_h[t] = \overline{W}_{h+1}[t] \qquad \forall h \in \mathcal{H} \tag{70b}$$

$$\implies s_h[t] \frac{W_1}{\overline{W}_h[t]} = s_{h+1}[t] \frac{W_1}{\overline{W}_{h+1}[t]} \qquad \forall h \in \mathcal{H} \tag{70c}$$

$$\implies s_h[t+1] = s_{h+1}[t+1] \qquad \forall h \in \mathcal{H} \tag{70d}$$

$$\implies \delta(h, h+1)[t+1] = 0 \qquad \forall h \in \mathcal{H} \tag{70e}$$

$$\implies \sum_{g=1}^{n-1} \|\delta(g, g+1)[t+1]\|_p = 0 \tag{70f}$$

Next, we prove the reverse direction (for any $p$):

$$\sum_{g=1}^{n-1} \|\delta(g, g+1)[t+1]\|_p = 0 \implies \delta(h, h+1)[t+1] = 0 \qquad \forall h \in \mathcal{H} \tag{71a}$$

$$\implies s_h[t+1] = s_{h+1}[t+1] \qquad \forall h \in \mathcal{H} \tag{71b}$$

$$\implies s_h[t]\frac{W_1}{\overline{W}_h[t]} = s_{h+1}[t]\frac{W_1}{\overline{W}_{h+1}[t]} \qquad \forall h \in \mathcal{H} \tag{71c}$$

$$\implies s_h[t] = s_{h+1}[t] \qquad \forall h \in \mathcal{H} \tag{71d}$$

$$\implies \delta(h, h+1)[t] = 0 \qquad \forall h \in \mathcal{H} \tag{71e}$$

$$\implies \sum_{g=1}^{n-1} \|\delta(g, g+1)[t]\|_p = 0 \tag{71f}$$

where Eq. (71d) follows from

$$\frac{s_g}{s_g W_1 + (1 - s_g)W_0} = \frac{s_h}{s_h W_1 + (1 - s_h)W_0} \tag{72a}$$

$$\implies s_g(s_h W_1 + (1 - s_h)W_0) = s_h(s_g W_1 + (1 - s_g)W_0) \tag{72b}$$

$$\implies W_0 s_g = W_0 s_h \tag{72c}$$

$$\implies s_g = s_h \tag{72d}$$

**Proof of Theorem 7**

**Theorem 7 Statement.**

Disregarding boundary states by Assumption 1, the replicator equation, Eq. (7), implies
$$\mathrm{sgn}\left(\bar{s}[t+1] - \bar{s}[t]\right) = \mathrm{sgn}\left(W_1(\phi[t]) - W_0(\phi[t])\right) \tag{73}$$

*Proof of Theorem 7.* There are three mutually exclusive cases we must consider by appeal to the replicator equation (Eq. (7)) and Assumption 1 ($s_g \in (0, 1)$ for all $g$ in $\mathcal{G}$). Specifically, we verify that

$$W_1 > W_0 \implies \forall g \in \mathcal{G}, \quad \frac{W_1}{\overline{W}_g} > 1 \implies \bar{s}[t+1] > \bar{s}[t] \tag{74a}$$

$$W_1 = W_0 \implies \forall g \in \mathcal{G}, \quad \frac{W_1}{\overline{W}_g} = 1 \implies \bar{s}[t+1] = \bar{s}[t] \tag{74b}$$

$$W_1 < W_0 \implies \forall g \in \mathcal{G}, \quad \frac{W_1}{\overline{W}_g} < 1 \implies \bar{s}[t+1] < \bar{s}[t] \tag{74c}$$

The sign of $W_1 - W_0$ therefore determines the sign of $\bar{s}[t+1] - \bar{s}[t]$ directly. Likewise, the sign of $\bar{s}[t+1] - \bar{s}[t]$ implies the sign of $W_1 - W_0$, for any discrepancy would imply a contradiction:

$$\bar{s}[t+1] > \bar{s}[t] \implies W_1 > W_0 \tag{75a}$$

$$\bar{s}[t+1] = \bar{s}[t] \implies W_1 = W_0 \tag{75b}$$

$$\bar{s}[t+1] < \bar{s}[t] \implies W_1 < W_0 \tag{75c}$$

**Proof of Theorem 8**

**Theorem 8 Statement.**

It is necessary and sufficient for a system at equilibrium that $W_1 = W_0$ or for the system to occupy some vertex of the state space.

$$\text{at equilibrium} \iff \begin{cases} W_1 = W_0 & \text{(internal equilibrium)} \\ \forall g \in \mathcal{G}, \quad s_g \in \{0, 1\} & \text{(trivial equilibrium)} \end{cases} \tag{76}$$

*Proof of Theorem 8.* By the definition of equilibrium (Definition 6), the forward direction implies equality between $s_g[t]$ and $s_g[t_0 + 1]$ for all $g \in \mathcal{G}$. It follows from the replicator equation (Eq. (7)) that at least one condition must be met to guarantee this equality, and at least one is consequent:

$$\forall g \in \mathcal{G}, \tag{77a}$$

$$s_g[t_0 + 1] = s_g[t_0] \iff \left( \frac{W_1}{W_1 s_g + W_0(1 - s_g)}[t_0] = 1 \ \lor \ s_g[t_0] = 0 \right) \tag{77b}$$

We consider the first case further and note that it is true if and only if $W_1 = W_0$ or $s_g = 1$.

$$\frac{W_1}{W_1 s_g + W_0(1 - s_g)} = 1 \iff W_1 = W_1 s_g + W_0(1 - s_g) \tag{78a}$$

$$\iff (1 - s_g)W_1 = W_0(1 - s_g) \tag{78b}$$

$$\iff \left( W_1 = W_0 \ \lor \ s_g = 1 \right) \tag{78c}$$

Thus the tree of cases terminates with, for each $g \in \mathcal{G}$, any one of $s_g = 0$, $s_g = 1$, or $W_0 = W_1$, as necessary and sufficient for the state to be at equilibrium. We note that if $W_0 = W_1$, no other conditions must be considered separately for different values of $g$. We may therefore re-express these conditions for equilibrium succinctly as the two cases we set out to show:

$$\text{at equilibrium} \iff \left( W_1 = W_0 \ \lor \ \forall g \in \mathcal{G}, \ s_g \in \{0, 1\} \right) \tag{79}$$

*Note: It is possible that $W_1 = W_0$ at some trivial equilibrium, and so our use of the term* internal *equilibrium does not strictly limit us to consideration of states in the* interior *of the state space (i.e., points removed from the boundary).*

## Proof of Theorem 9

**Theorem 9 Statement.** $W_1(\phi) - W_0(\phi)$ is strictly quasi-concave in $\phi$. This guarantees that no more than two zeros of the function $W_1 - W_0$ exist.

*Proof of Theorem 9.* We proceed by characterizing the function $W_1(\phi) - W_0(\phi)$, starting with its zeros. By Theorem 2 and Lemma 2.1, the values of $\phi$ for which $W_1(\phi) - W_0(\phi) = 0$ must satisfy

$$\left( U_{1\hat{\imath}} + (U_{1\hat{0}} - U_{1\hat{\imath}})Q_1(\phi) \right) - \left( U_{0\hat{\imath}} + (U_{0\hat{0}} - U_{0\hat{\imath}})Q_0(\phi) \right) = 0 \tag{80}$$

Next, we consider the first derivative of $W_1 - W_0$ with respect to $\phi$:

$$\frac{d}{d\phi}\left( W_1(\phi) - W_0(\phi) \right) = q_1(\phi)(U_{1\hat{0}} - U_{1\hat{\imath}}) - q_0(\phi)(U_{0\hat{0}} - U_{0\hat{\imath}}) \tag{81a}$$

$$= \left( \frac{q_1(\phi)}{q_0(\phi)} - \frac{U_{0\hat{0}} - U_{0\hat{\imath}}}{U_{1\hat{0}} - U_{1\hat{\imath}}} \right) \left( q_0(\phi)(U_{1\hat{0}} - U_{1\hat{\imath}}) \right) \tag{81b}$$

Recall that $U_{i\hat{0}} \neq U_{i\hat{\imath}}$ and $U_{1\hat{0}} < U_{1\hat{\imath}}$ (Assumption 9). By the strict (increasing) monotonicity of $\frac{q_1(\phi)}{q_0(\phi)}$ in $\phi$ and strict positivity of $q_0(\phi)$, both guaranteed by Assumption 3, the sign of this expression can change at most once as $\phi$ is varied from $-\infty$ to $\infty$. We denote the value of $\phi$ at which the sign of this first derivative changes as $\phi^\star$:

$$\frac{q_1(\phi^\star)}{q_0(\phi^\star)} = \frac{U_{0\hat{0}} - U_{0\hat{\imath}}}{U_{1\hat{0}} - U_{1\hat{\imath}}} \tag{82}$$

Moreover, it follows that

$$\phi < \phi^\star \implies \frac{d}{d\phi}W_1 - W_0 > 0 \tag{83a}$$

$$\phi > \phi^\star \implies \frac{d}{d\phi}W_1 - W_0 < 0 \tag{83b}$$

$W_1 - W_0$ is therefore strictly quasi-concave, from which it follows that only two zeros of the function can exist (*By contradiction, more than two zeros would require the function, which has no discontinuities, to invert its slope more than once.*)

For completeness, we may also take a second derivative of $W_1 - W_0$ with respect to $\phi$:

$$\frac{d^2}{d\phi^2}\Big(W_1(\phi) - W_0(\phi)\Big) = \frac{d}{d\phi}\left(\frac{q_1(\phi)}{q_0(\phi)}\right)\Big(q_0(\phi)(U_{1\hat{0}} - U_{1\hat{1}})\Big)$$
$$+ \left(\frac{q_1(\phi)}{q_0(\phi)} - \frac{U_{0\hat{0}} - U_{0\hat{1}}}{U_{1\hat{0}} - U_{1\hat{1}}}\right)\left(\frac{d}{d\phi}q_0(\phi)\right) \tag{84}$$

Doing so, we observe that $W_1 - W_0$ may have any number of inflection points, but $\phi^\star$ cannot be one of them. We see this because the second term of the expression above evaluated at $\phi^\star$ must be zero, but the first term must be non-zero by Assumption 3 and Assumption 9. It follows that $\phi^\star$ is the unique occurrence of a local extremum and therefore a global extremum of $W_1 - W_0$.

## Proof of Theorem 10

**Theorem 10 Statement.** If the state of the system asymptotically approaches an internal equilibrium, the nullity of $p$-total qualification rate disparity is preserved in the limit of infinite time.

$$p \geq 1; \quad \lim_{t'\to\infty}(W_1 - W_0) = 0 \implies \Big(\big\|D[t]\big\|_p = 0 \iff \lim_{t'\to\infty}\big\|D[t']\big\|_p = 0\Big) \tag{85}$$

*Proof of Theorem 10.* Assuming that a state asymptotically approaches the equilibrium hyperplane $(\lim_{t'\to\infty}(W_1-W_0) = 0)$, Let us first prove the forward direction of the desired mutual implication. If a state starts with zero total qualfication rate disparity, it follows by Theorem 5 that the total qualification rate disparity remains zero for all time, and so

$$p \geq 1; \quad \lim_{t'\to\infty}(W_1 - W_0) = 0 \implies \Big(\big\|D[t]\big\|_p = 0 \implies \lim_{t'\to\infty}\big\|D[t']\big\|_p = 0\Big) \tag{86}$$

For the reverse direction, let us define $\mathbf{s}^\star$ as the unique disparity-free state on the stable internal equilibrium hyperplane, such that

$$\big\|D(\mathbf{s}^\star)\big\|_p = 0 \tag{87}$$

We may phase the assumptions $\lim_{t'\to\infty}(W_1 - W_0) = 0$ and $\lim_{t'\to\infty}\|D[t']\|_p = 0$ jointly as the condition

$$\lim_{t'\to\infty}\mathbf{s} = \mathbf{s}^\star \tag{88}$$

By the Weierstrass definition of a limit, this is

$$\forall \varepsilon > 0, \exists t_0, \forall t > t_0, \quad \|\mathbf{s} - \mathbf{s}^\star\|_p < \varepsilon. \tag{89}$$

For any $\varepsilon$, we have thus assumed that there exists some time $t_0$ beyond which $\mathbf{s}$ is within $\varepsilon$ of $\mathbf{s}^\star$. In particular, we are free to choose $\varepsilon$ small enough that the local dynamics of the system are well approximated by the linearization undertaken in the proof of Theorem 11:

Because the system is well approximated to first order within any sufficiently-small $\varepsilon$-neighborhood of the equilibrium hyperplane, the preimage of $\mathbf{s}^\star$ in the infinite-time limit within this neighborhood lies along the line through $\mathbf{s}^\star$ parallel to the *sole* eigenvector of the Jacobian with non-zero eigenvalue: $\mathbf{v}$.

When $\forall g, \delta(g, g + 1) = 0$, $\mathbf{v}$ is orthogonal to the internal equilibrium hyperplane (Eq. (21)), therefore, all states in the preimage of $\mathbf{s}^\star$ also satisfy $\forall g, \delta(g, g + 1) = 0$ and exhibit zero $p$-total qualification rate disparity. We may then appeal to induction and Theorem 5 to note that the entire trajectory of of the state must have had zero total disparity.

$$p \geq 1; \quad \lim_{t'\to\infty}(W_1 - W_0) = 0 \implies \Big(\big\|D[t]\big\|_p = 0 \impliedby \lim_{t'\to\infty}\big\|D[t']\big\|_p = 0\Big) \tag{90}$$

This completes the proof.

## A Series of Lemmas for Linear Stability Analysis

**Lemma 11.1.**

$$\frac{\partial}{\partial\phi}\frac{W_1}{W_g}\bigg|_{\text{eq}} = \frac{1}{W_{\text{eq}}}(1 - s_g)\frac{\partial}{\partial\phi}(W_1 - W_0)\bigg|_{\text{eq}} \tag{91}$$

*Proof of Lemma 11.1.* We directly differentiate the expression evaluated at equilibrium, recalling that $W_{\text{eq}} = W_0|_{\text{eq}} = W_1|_{\text{eq}} = \overline{W}_g|_{\text{eq}} \; \forall g \in \mathcal{G}$ and $W_g = s_g W_1 + (1 - s_g) W_0$.

$$\frac{\partial}{\partial \phi} \frac{W_1}{\overline{W}_g}\bigg|_{\text{eq}} = \frac{1}{W_{\text{eq}}} \frac{\partial W_1}{\partial \phi}\bigg|_{\text{eq}} - \frac{1}{W_{\text{eq}}} \frac{\partial \overline{W}_g}{\partial \phi}\bigg|_{\text{eq}} \tag{92a}$$

$$= \frac{1}{W_{\text{eq}}} \frac{\partial}{\partial \phi} \Big( W_1 - s_g W_1 - (1 - s_g) W_0 \Big)\bigg|_{\text{eq}} \tag{92b}$$

$$= \frac{1}{W_{\text{eq}}} (1 - s_g) \frac{\partial}{\partial \phi} (W_1 - W_0)\bigg|_{\text{eq}} \tag{92c}$$

**Lemma 11.2.** $\forall g \in \mathcal{G}$,

$$\frac{\partial \overline{s}}{\partial s_g} = \mu_g, \quad \frac{\partial \delta(g, g+1)}{\partial s_g} = 1, \quad \frac{\partial \delta(g-1, g)}{\partial s_g} = -1 \tag{93}$$

| *Proof of Lemma 11.2.* By Eq. (2) and Eq. (10), the result is immediate.

**Lemma 11.3.**

$$\frac{\partial s_g}{\partial \overline{s}} = 1 \tag{94}$$

| *Proof of Lemma 11.3.* By Eq. (12), the result is immediate.

**Lemma 11.4.**

$$\frac{\partial s_g}{\partial \delta(g, h)} = \begin{cases} 1 - \mu_1 - \mu_2 - \ldots - \mu_g & h = g + 1 \\ \mu_1 + \mu_2 + \ldots + \mu_{g-1} & h = g - 1 \end{cases} = \begin{cases} 1 + \beta_g & h = g + 1 \\ -\mu_g - \beta_g & h = g - 1 \end{cases} \tag{95}$$

$$\frac{\partial s_h}{\partial \delta(g, h)} = \begin{cases} -\mu_1 - \mu_2 - \ldots - \mu_g & h = g + 1 \\ -1 + \mu_1 + \mu_2 + \ldots + \mu_{g-1} & h = g - 1 \end{cases} = \begin{cases} \beta_g & h = g + 1 \\ -\mu_g - \beta_g - 1 & h = g - 1 \end{cases} \tag{96}$$

| *Proof of Lemma 11.4.* The result follows from Eq. (12), noting $\delta(g, h) = -\delta(h, g)$.

**Lemma 11.5.** *Taking a partial derivative with respect to $s_g$ while holding all other $s_h, h \neq g$ fixed,*

$$\frac{\partial}{\partial s_g[t]} s_g[t+1]\bigg|_{\text{eq}} = 1 + \mu_g \frac{1}{W_{\text{eq}}} \left( \frac{\partial \phi}{\partial \overline{s}} \right) s_g (1 - s_g) \frac{\partial}{\partial \phi} (W_1 - W_0)\bigg|_{\text{eq}} \tag{97}$$

*Holding $\phi$ constant as well,*

$$\left( \frac{\partial}{\partial s_g[t]} \right)_\phi s_g[t+1]\bigg|_{\text{eq}} = 1 \tag{98}$$

*When $\phi$ is held constant when taking a partial derivative with respect to $s_g$, we shall denote the partial derivative with $\phi$ in the subscript, as in the equation above, and omit this subscript otherwise.*

| *Proof of Lemma 11.5.* Let us begin by proving the second equality, observing first that

$$\frac{\partial}{\partial s_g} \overline{W}_g = \frac{\partial}{\partial s_g} \Big( s_g W_1 + (1 - s_g) W_0 \Big) = W_1 - W_0 \tag{99}$$

With $\phi$ fixed, $W_1$ does not depend on $s_g$. Therefore, substituting $s_g[t+1] = s_g \frac{W_1}{W_g}$,

$$\left(\frac{\partial}{\partial s_g}\right)_\phi \left(s_g \frac{W_1}{\overline{W}_g}\right)\Bigg|_{\text{eq}} = \frac{W_1}{\overline{W}_g}\Bigg|_{\text{eq}} - s_g \frac{W_1}{\overline{W}_g^2}(W_1 - W_0)\Bigg|_{\text{eq}} \tag{100a}$$

$$= \frac{W_{\text{eq}}}{W_{\text{eq}}} - 0 \tag{100b}$$

$$= 1 \tag{100c}$$

We next address the first equality. By Lemma 11.2,

$$\left(\frac{\partial \phi}{\partial s_g}\right) = \left(\frac{\partial \overline{s}}{\partial s_g}\right)\left(\frac{\partial \phi}{\partial \overline{s}}\right) = \mu_g \left(\frac{\partial \phi}{\partial \overline{s}}\right) \tag{101}$$

By Lemma 11.1,

$$\left(\frac{\partial}{\partial s_g}\right)\left(s_g \frac{W_1}{\overline{W}_g}\right)\Bigg|_{\text{eq}} = \left(\frac{\partial}{\partial s_g}\right)_\phi \left(s_g \frac{W_1}{\overline{W}_g}\right)\Bigg|_{\text{eq}} + s_g \left(\frac{\partial \phi}{\partial s_g}\right)\left(\frac{\partial}{\partial \phi} \frac{W_1}{\overline{W}_g}\right)\Bigg|_{\text{eq}} \tag{102a}$$

$$= 1 + \mu_g \frac{s_g}{W_{\text{eq}}}\left(\frac{\partial \phi}{\partial \overline{s}}\right)(1 - s_g)\frac{\partial}{\partial \phi}(W_1 - W_0)\Bigg|_{\text{eq}} \tag{102b}$$

**Fact 11.1.** *We note when differentiating an expression $\mathfrak{g}$ with respect to an expression $\mathfrak{f}$, each involving each $s_g$ and $\phi$ (which depends on each $s_g$), we may invoke the chain rule to treat $\phi$ as an independent function input from the beginning, or we may treat the effect on $\phi$ due to perturbation of each $s_g$ separately. It is for this reason that we have been explicit about which variables are fixed in the partial derivatives of Lemma 11.5.*

$$\frac{\partial}{\partial s_g}\mathfrak{f}(s_g, \phi) = \frac{\partial \phi}{\partial s_g}\frac{\partial \mathfrak{f}}{\partial \phi} + \left(\frac{\partial \mathfrak{f}}{\partial s_g}\right)_\phi \tag{103}$$

$$\frac{\partial \phi}{\partial \mathfrak{f}} = \sum_{g \in \mathcal{G}} \frac{\partial s_g}{\partial \mathfrak{f}}\frac{\partial \phi}{\partial s_g} \tag{104}$$

*Therefore,*

$$\frac{\partial}{\partial \mathfrak{f}}\mathfrak{g}(s_g, \phi) = \sum_{g \in \mathcal{G}}\left(\frac{\partial s_g}{\partial \mathfrak{f}}\right)\frac{\partial}{\partial s_g}\mathfrak{g}(s_g, \phi) \tag{105a}$$

$$= \sum_{g \in \mathcal{G}}\left(\frac{\partial s_g}{\partial \mathfrak{f}}\right)\left(\frac{\partial \phi}{\partial s_g}\frac{\partial \mathfrak{g}}{\partial \phi} + \left(\frac{\partial \mathfrak{g}}{\partial s_g}\right)_\phi\right) \tag{105b}$$

$$= \sum_{g \in \mathcal{G}}\left(\frac{\partial s_g}{\partial \mathfrak{f}}\frac{\partial \phi}{\partial s_g}\right)\frac{\partial \mathfrak{g}}{\partial \phi} + \sum_{g \in \mathcal{G}}\left(\frac{\partial s_g}{\partial \mathfrak{f}}\right)\left(\frac{\partial \mathfrak{g}}{\partial s_g}\right)_\phi \tag{105c}$$

$$= \frac{\partial \phi}{\partial \mathfrak{f}}\frac{\partial \mathfrak{g}}{\partial \phi} + \sum_{g \in \mathcal{G}}\left(\frac{\partial s_g}{\partial \mathfrak{f}}\right)\left(\frac{\partial \mathfrak{g}}{\partial s_g}\right)_\phi \tag{105d}$$

*For convenience, we will treat $\phi$ as an independent function input (i.e., we will invoke the chain rule as in Eq. (105d)) when proving Lemma 11.6, Lemma 11.7, and Lemma 11.8.*

**Lemma 11.6.**

$$\frac{\partial(\overline{s}[t+1] - \overline{s}[t])}{\partial \overline{s}[t]}\Bigg|_{\text{eq}} = \frac{1}{W_{\text{eq}}}\left(\frac{\partial \phi}{\partial \overline{s}}\right)\left(\sum_{g \in \mathcal{G}}\mu_g s_g(1 - s_g)\right)\frac{\partial}{\partial \phi}(W_1 - W_0)\Bigg|_{\text{eq}} \tag{106}$$

*Proof of Lemma 11.6.* Noting that $\overline{s}[t+1]$ depends on each $s_g$ and $\phi$,

$$\overline{s}[t+1] = \sum_{g \in \mathcal{G}} \mu_g s_g \frac{W_1(\phi)}{\overline{W}_g(\phi)} \tag{107}$$

we may use the chain rule (Fact 11.1),

$$\frac{\partial}{\partial \overline{s}} f(s_1, s_2, ..., s_n, \phi) = \sum_{g \in \mathcal{G}} \frac{\partial s_g}{\partial \overline{s}} \left( \frac{\partial f}{\partial s_g} \right)_\phi + \frac{\partial \phi}{\partial \overline{s}} \frac{\partial f}{\partial \phi} \tag{108}$$

to compute, referencing Lemma 11.1, Lemma 11.2, Lemma 11.3, and Lemma 11.5,

$$\left. \frac{\partial(\overline{s}[t+1] - \overline{s}[t])}{\partial \overline{s}[t]} \right|_{\text{eq}} = \left. \frac{\partial \overline{s}[t+1]}{\partial \overline{s}[t]} - 1 \right|_{\text{eq}} \tag{109a}$$

$$= \left( \sum_{g \in \mathcal{G}} \left( \frac{\partial s_g}{\partial \overline{s}} \right) \left( \frac{\partial}{\partial s_g} \right)_\phi \left( \mu_g s_g \frac{W_1}{\overline{W}_g} \right) + \left( \frac{\partial \phi}{\partial \overline{s}} \right) \left( \frac{\partial}{\partial \phi} \sum_{g \in \mathcal{G}} \mu_g s_g \frac{W_1}{\overline{W}_g} \right) - 1 \right) \bigg|_{\text{eq}} \tag{109b}$$

$$= \sum_{g \in \mathcal{G}} \mu_g + \left. \left( \frac{\partial \phi}{\partial \overline{s}} \right) \frac{1}{W_{\text{eq}}} \sum_{g \in \mathcal{G}} \mu_g s_g (1 - s_g) \frac{\partial}{\partial \phi} (W_1 - W_0) \right|_{\text{eq}} - 1 \tag{109c}$$

$$= \left. \frac{1}{W_{\text{eq}}} \left( \frac{\partial \phi}{\partial \overline{s}} \right) \left( \sum_{g \in \mathcal{G}} \mu_g s_g (1 - s_g) \right) \frac{\partial}{\partial \phi} (W_1 - W_0) \right|_{\text{eq}} \tag{109d}$$

**Lemma 11.7.**

$$\left. \frac{\partial}{\partial \overline{s}[t]} \left( \delta(g, h)[t+1] - \delta(g, h)[t] \right) \right|_{\text{eq}} \tag{110a}$$

$$= \left. \frac{1}{W_{\text{eq}}} \left( \frac{\partial \phi}{\partial \overline{s}} \right) \delta(g, h)(1 - s_g - s_h) \frac{\partial}{\partial \phi} (W_1 - W_0) \right|_{\text{eq}} \tag{110b}$$

*Proof of Lemma 11.7.* Since $\overline{s}$ and $\delta(g, h)$ are independent coordinates, the partial derivative of one with respect to the other at the same time is identically zero.

$$\frac{\partial}{\partial \overline{s}} \delta(g, h) = 0 \tag{111}$$

The left hand side of the target equality is therefore equal to

$$\left. \frac{\partial}{\partial \overline{s}[t]} \delta(g, h)[t+1] \right|_{\text{eq}}$$

From the chain rule (Fact 11.1), Lemma 11.1, Lemma 11.3, and Lemma 11.5, it follows that

$$\frac{\partial}{\partial \overline{s}[t]} \delta(g,h)[t+1]\Big|_{\text{eq}} \tag{112a}$$

$$= \left( \left( \frac{\partial \phi}{\partial \overline{s}} \right) \frac{\partial \delta(g,h)[t+1]}{\partial \phi} + \frac{\partial s_g}{\partial \overline{s}} \left( \frac{\partial s_g[t+1]}{\partial s_g[t]} \right)_\phi - \frac{\partial s_h}{\partial \overline{s}} \left( \frac{\partial s_h[t+1]}{\partial s_h[t]} \right)_\phi \right) \Big|_{\text{eq}} \tag{112b}$$

$$= \left( \frac{\partial \phi}{\partial \overline{s}} \right) \frac{\partial}{\partial \phi} \delta(g,h)[t+1]\Big|_{\text{eq}} \tag{112c}$$

$$= \left( \frac{\partial \phi}{\partial \overline{s}} \right) \frac{\partial}{\partial \phi} \left( s_g \frac{W_1}{\overline{W}_g} - s_h \frac{W_1}{\overline{W}_h} \right) \Big|_{\text{eq}} \tag{112d}$$

$$= \frac{1}{W_{\text{eq}}} \left( \frac{\partial \phi}{\partial \overline{s}} \right) \left( s_g(1-s_g) - s_h(1-s_h) \right) \frac{\partial}{\partial \phi} (W_1 - W_0)\Big|_{\text{eq}} \tag{112e}$$

$$= \frac{1}{W_{\text{eq}}} \left( \frac{\partial \phi}{\partial \overline{s}} \right) \delta(g,h)(1-s_g-s_h) \frac{\partial}{\partial \phi} (W_1 - W_0)\Big|_{\text{eq}} \tag{112f}$$

**Lemma 11.8.**

$$\left( \frac{\partial \overline{s}[t+1] - \overline{s}[t]}{\partial \delta(g,h)[t]} \right) \Big|_{\text{eq}} = 0 \tag{113}$$

$$\left( \frac{\partial \delta(g,h)[t+1] - \delta(g,h)[t]}{\partial \delta(g,h)[t]} \right) \Big|_{\text{eq}} = 0 \tag{114}$$

$$\left( \frac{\partial \delta(h,h+1)[t+1] - \delta(h,h+1)[t]}{\partial \delta(g,g+1)[t]} \right) \Big|_{\text{eq}} = 0, \quad \forall h \neq g \tag{115}$$

*Proof of Lemma 11.8.* By Theorem 1, $\phi$ depends only on $\overline{s}$, which is held constant during partial differentiation by $\delta(g,h)$. Therefore,

$$\frac{\partial \phi}{\partial \delta(g,h)} = 0 \tag{116}$$

Consider any expression $\mathfrak{f}$ which depends *linearly* on each $s_g[t+1]$.

$$\mathfrak{f}(s_g : g \in \mathcal{G}) := \sum_{g \in \mathcal{G}} f_g s_g, \quad f_g \in \mathbf{R} \tag{117}$$

where we introduce a "vector builder" notation $\mathbf{s} = \left( s_g : g \in \mathcal{G} \right)$ for brevity. We may use the linearity of differentiation to concisely deal with derivatives of linear combinations of $\mathfrak{f}$. We consider will expressions without explicit time dependence to correspond to time $[t]$. By the chain rule (Fact 11.1), Eq. (116), and Lemma 11.5,

$$\frac{\partial}{\partial \delta(g,h)} \left( \mathfrak{f}[t+1] - \mathfrak{f}[t] \right) \Big|_{\text{eq}} \tag{118a}$$

$$= \left( \sum_{i \in \mathcal{G}} \frac{\partial s_i}{\partial \delta(g,h)} \left( \frac{\partial}{\partial s_i} \right)_\phi + \frac{\partial \phi}{\partial \delta(g,h)} \frac{\partial}{\partial \phi} \right) \mathfrak{f}\left( s_i[t+1] - s_i[t] : i \in \mathcal{G} \right) \Big|_{\text{eq}} \tag{118b}$$

$$= \mathfrak{f}\left( \sum_{i \in \mathcal{G}} \frac{\partial s_i}{\partial \delta(g,h)} \left( \frac{\partial s_i[t+1]}{\partial s_i[t]} - \frac{\partial s_i[t]}{\partial s_i[t]} \right) : i \in \mathcal{G} \right) \tag{118c}$$

$$= \mathfrak{f}(0 : i \in \mathcal{G}) = 0 \tag{118d}$$

We conclude that perturbing any $\delta$ while holding $\overline{s}$ constant has no effect on the evolution of dynamical variables that are linear in $\mathbf{s}$ at equilibrium. This includes each $\delta$ and $\overline{s}$.

## Proof of Theorem 11

**Theorem 11 Statement.** The Jacobian $J$ simplifies to a scalar multiplied by a matrix with a single non-zero column $\mathbf{v}$ in the last position.

$$J\bigg|_{\text{eq}} = \frac{1}{W_{\text{eq}}}\left(\frac{d\phi}{d\bar{s}}\right)\left(\frac{d}{d\phi}(W_1 - W_0)\right)\left[\mathbf{0}^{(n \times n-1)}\bigg|\mathbf{v}\right], \quad \mathbf{v} := \begin{bmatrix} \delta(1,2)(1-s_1-s_2) \\ \delta(2,3)(1-s_2-s_3) \\ \cdots \\ \delta(n-1,n)(1-s_{n-1}-s_n) \\ \sum_{g \in \mathcal{G}} \mu_g s_g(1-s_g) \end{bmatrix} \quad (119)$$

*Proof of Theorem 11.* The zero entries in the Jacobian matrix are a consequence of Lemma 11.8. Lemma 11.7 and Lemma 11.6 provide us with the last column of the matrix $J$ in the desired form:

$$\frac{\partial \delta(g,h)[t+1] - \delta(g,h)[t]}{\partial \bar{s}[t]}\bigg|_{\text{eq}} = \frac{1}{W_{\text{eq}}}\left(\frac{\partial \phi}{\partial \bar{s}}\right)\delta(g,h)(1 - s_g - s_h)\frac{\partial}{\partial \phi}(W_1 - W_0)\bigg|_{\text{eq}} \quad (120)$$

$$\frac{\partial(\bar{s}[t+1] - \bar{s}[t])}{\partial \bar{s}[t]}\bigg|_{\text{eq}} = \frac{1}{W_{\text{eq}}}\left(\frac{\partial \phi}{\partial \bar{s}}\right)\left(\sum_{g \in \mathcal{G}} \mu_g s_g(1 - s_g)\right)\frac{\partial}{\partial \phi}(W_1 - W_0)\bigg|_{\text{eq}} \quad (121)$$

## Proof of Corollary 11.1

**Corollary 11.1 Statement.**

At equilibrium, any state displacement vector with zero $\bar{s}$ component is an eigenvector of $J$ with eigenvalue 0, while $\mathbf{v}$ is an eigenvector of $J$ with eigenvalue $\lambda$:

$$\lambda := \left(\sum_{g \in \mathcal{G}} \mu_g s_g(1 - s_g)\right)\frac{1}{W_{\text{eq}}}\left(\frac{d\phi}{d\bar{s}}\right)\left(\frac{d}{d\phi}(W_1 - W_0)\right)\bigg|_{\text{eq}} \quad (122)$$

*Proof of Corollary 11.1.* **Corollary 11.1** follows by inspection of $J$ in Eq. (119).

## Proof of Corollary 11.2

**Corollary 11.2 Statement.** As a consequence of Corollary 1.1, which states $\frac{d\phi}{d\bar{s}} < 0$, the eigenvalue $\lambda$ in Eq. (22) is negative, (and the associated equilibrium hyperplane stable) iff $\frac{d}{d\phi}(W_1 - W_0)|_{\text{eq}} > 0$. This prescribes precisely the value $\phi^+$ for the stable equilibrium hyperplane.

*Proof of Corollary 11.2.* This is a consequence of Corollary 1.1 and Corollary 11.1 given Assumption 1 (each $s_g$ is interior) and the restriction of $W_y \in [0, \infty)$ as specified in the replicator equation (Eq. (7)). The eigenvalue $\lambda$ is negative, (and the associated equilibrium hyperplane stable) iff

$$\frac{d}{d\phi}(W_1 - W_0)\bigg|_{\text{eq}} > 0 \quad (123)$$

This prescribes precisely the value $\phi^+$ for the stable equilibrium hyperplane.

## Proof of Theorem 13

**Theorem 13 Statement.** For policies defined by group-specific thresholds $\phi_g$, the equivalence of these feature thresholds ($\forall g, \phi_g = \phi$) is necessary and sufficient to satisfy Equalized Odds given the group-independence of each $q_y$ (Assumption 2).

*Proof of Theorem 13.* The forward direction (group-independence satisfies Equalized Odds) follows from the group-independence of $Q_y$ (Definition 2.1). The reverse direction follows from the same; specifically, as functions of $\phi$,

$$\Pr(\hat{Y} = 0 \mid Y = y) = Q_y(\phi) \quad (124)$$

$$\Pr(\hat{Y} = 1 \mid Y = y) = (1 - Q_y(\phi)) \quad (125)$$

are each monotonic, and any specified value of $\Pr(\hat{Y} = \hat{y} \mid Y = y)$ corresponds to a unique $\phi$ value that must be shared by all groups.

## Proof of Corollary 13.1

**Corollary 13.1 Statement.** Equalized Odds does not imply long-term fairness in our model.

*Proof of Corollary 13.1.* By contradiction, we have shown that a group-independent threshold policy satisfies Equalized Odds (Theorem 13), yet long-term fairness is violated by persistent qualification rate disparities (Theorem 10).

## Proof of Theorem 14

**Theorem 14 Statement.** $\Theta(\epsilon)$ perturbations of a group-independent $\phi$ at internal equilibrium induce motion, which, to first-order approximation (*i.e.*, ignoring $\mathcal{O}(\epsilon^2)$ terms), is parallel to the eigenvector $\mathbf{v}$.

*Proof of Theorem 14.* We note that the a perturbation to $\phi$ at internal equilibrium causes a change in state vector parallel to the eigenvector $\mathbf{v}$, where

$$
\mathbf{v} = \frac{\partial \mathbf{r}}{\partial \overline{s}} = \begin{bmatrix} \delta(1,2)(1-s_1-s_2) \\ \delta(2,3)(1-s_2-s_3) \\ ... \\ \delta(n-1,n)(1-s_{n-1}-s_n) \\ \sum_{g \in \mathcal{G}} \mu_g s_g(1-s_g) \end{bmatrix}
\tag{126}
$$

By use of the chain rule with Lemma 11.7, or direct application of Lemma 11.1, we note

$$
\frac{\partial}{\partial \phi}\left(\delta(g,h)[t+1] - \delta(g,h)[t]\right)\bigg|_{\text{eq}} = \left(\frac{\partial \phi}{\partial \overline{s}}\right)^{-1} \frac{\partial \delta(g,h)[t+1] - \delta(g,h)[t]}{\partial \overline{s}[t]}\bigg|_{\text{eq}}
\tag{127a}
$$

$$
= \frac{1}{W_{\text{eq}}} \frac{\partial}{\partial \phi}(W_1 - W_0)\bigg|_{\text{eq}} \left(s_g(1-s_g) - s_h(1-s_h)\right)\bigg|_{\text{eq}}
\tag{127b}
$$

$$
= \frac{1}{W_{\text{eq}}} \frac{\partial}{\partial \phi}(W_1 - W_0)\bigg|_{\text{eq}} \delta(g,h)(1-s_g-s_h)
\tag{127c}
$$

Likewise, pairing the chain rule with Lemma 11.6 or directly applying Lemma 11.1, we note

$$
\frac{\partial}{\partial \phi}\left(\overline{s}[t+1] - \overline{s}[t]\right)\bigg|_{\text{eq}} = \left(\frac{\partial \phi}{\partial \overline{s}}\right)^{-1} \frac{\partial(\overline{s}[t+1] - \overline{s}[t])}{\partial \overline{s}[t]}\bigg|_{\text{eq}}
\tag{128a}
$$

$$
= \frac{1}{W_{\text{eq}}} \frac{\partial}{\partial \phi}(W_1 - W_0)\bigg|_{\text{eq}} \sum_g \mu_g s_g(1-s_g)
\tag{128b}
$$

Together, our observations imply

$$
\frac{\partial}{\partial \phi}(\mathbf{s}[t+1] - \mathbf{s}[t])\bigg|_{\text{eq}} = \frac{1}{W_{\text{eq}}} \frac{\partial}{\partial \phi}(W_1 - W_0)\bigg|_{\text{eq}} \mathbf{v}
\tag{129}
$$

and perturbation of $\phi$ induces motion parallel to $\mathbf{v}$. For readers familiar with gradient descent but new to linear stability analysis, we offer the intuition that $\mathbf{v}$ is parallel to the gradient of $\phi$ in state space.

## Proof of Theorem 17

**Theorem 17 Statement.** Demographic parity requires sign-heterogeneous, group-dependent changes to the Laissez-Faire values of $\phi_g$ when $\pi$ is non-trivial (does not uniformly accept (reject)).

*Proof of Theorem 17.* The policy adopted by a classifier subject to demographic parity is given by

$$\pi = \underset{\pi}{\operatorname{argmax}} \sum_{y,\hat{y}=0}^{1} V_{y\hat{y}} \underset{\hat{Y}=\pi(X)}{\operatorname{Pr}} (Y = y, \hat{Y} = \hat{y}) \tag{130}$$

$$\text{subject to} \quad \operatorname{Pr}(\hat{Y} = 1 \mid G = g) = \operatorname{Pr}(\hat{Y} = 1 \mid G = h) \quad \forall g, h \in \mathcal{G}$$

Without allowing group-dependent values of $\phi$, the only solutions to $\pi$ when groups have differing qualification rates are the trivial policies $\pi = 0$ and $\pi = 1$. We therefore consider a solution that permits group-dependent thresholds $\phi_g$. We solve for these thresholds using the method of Lagrange multipliers. In the $s_g$ state basis, this requires that we satisfy, for Lagrange multipliers $L_h \in (-\infty, \infty)$, $h \in \{1, 2, ... n - 1\}$, the set of equations

$$\nabla_{\phi} u = \nabla_{\phi} \left( \sum_{h=1}^{n-1} L_h c_h \right) \tag{131}$$

where $\nabla_{\phi}$ denotes the vector operator such that the $g$th component is the partial derivative with respect to $\phi_g$; $u$ is the utility to be maximized; and each $c_h = 0$ represents a pairwise constraint between the probabilities of accepting an agent from two different groups ($h$ and $h + 1$).

$$u := \sum_{y,\hat{y}=0}^{1} V_{y\hat{y}} \underset{\hat{Y}=\pi(X)}{\operatorname{Pr}} (Y = y, \hat{Y} = \hat{y}) \tag{132a}$$

$$= \sum_{g} \mu_g \begin{pmatrix} V_{0\hat{0}}(1 - s_g)Q_0(\phi_g) \\ +V_{0\hat{1}}(1 - s_g)(1 - Q_0(\phi_g)) \\ +V_{1\hat{0}}s_g Q_1(\phi_g) \\ +V_{1\hat{1}}s_g(1 - Q_1(\phi_g)) \end{pmatrix} \tag{132b}$$

$$c_h := \begin{pmatrix} s_h Q_1(\phi_h) + (1 - s_h)Q_0(\phi_h) \\ -s_{h+1}Q_1(\phi_{h+1}) - (1 - s_{h+1})Q_0(\phi_{h+1}) \end{pmatrix} \tag{133}$$

Defining $L_0 = L_n = 0$ for notational convenience, Eq. (131) simplifies to a set $n$ equations indexed by $g \in \{1, 2, ..., n\}$

$$\mu_g \Big( (V_{0\hat{0}} - V_{0\hat{1}})q_0(\phi_g)(1 - s_g) + (V_{1\hat{0}} - V_{1\hat{1}})q_1(\phi_g)s_g \Big) \tag{134a}$$

$$= (L_g - L_{g-1}) \Big( s_g q_1(\phi_g) + (1 - s_g)q_0(\phi_g) \Big) \tag{134b}$$

From which the perturbed values $\phi_g$ may be derived:

$$\frac{q_1(\phi_g)}{q_0(\phi_g)} = \left( \frac{V_{0\hat{0}} - V_{0\hat{1}} - \gamma_g}{V_{1\hat{1}} - V_{1\hat{0}} + \gamma_g} \right) \left( \frac{1 - s_g}{s_g} \right) \tag{135a}$$

$$\gamma_g := \frac{L_g - L_{g-1}}{\mu_g}; \quad L_g = \frac{\partial u}{\partial c_g} \tag{135b}$$

We compare this equation with Eq. (45), noting that when each $\gamma_g = 0$ (*i.e.*, requiring that constraints $c_g$ are not active at locally optimal utility $u$), we recover a Laissez-fair policy:

$$\frac{q_1(\phi)}{q_0(\phi)} = \left( \frac{V_{0\hat{0}} - V_{0\hat{1}}}{V_{1\hat{1}} - V_{1\hat{0}}} \right) \left( \frac{1 - s_g}{s_g} \right) \tag{136}$$

For interpretation of the Lagrange multipliers $L_g$, also known as the *dual variables*, we refer the reader to Boyd et al. [43]. By the monotonicity of $q_1/q_0$, the effect of $\gamma_g$ in determining $\phi_g$ is therefore a perturbation to $\phi$, the sign of which is inverted relative to the sign of $\gamma_g$. Finally, having defined $L_0 = L_n = 0$, as a telescoping sum,

$$\sum_{g=1}^{n} (L_g - L_{g-1}) = 0 \tag{137}$$

Therefore,

$$\sum_{g=1}^{n} \mu_g \gamma_g = 0 \tag{138}$$

This guarantees in turn that set of group-specific *changes* to the group-specific values $\phi_g$ defined by a Laissez-Fair policy (Eq. (136)) must be sign-heterogeneous to satisfy Demographic Parity.

We comment that a solution for each $\gamma_g$ that satisfying constraints $c_g$ requires the solution of differential equation(s) in $q_y$ (*i.e.*, equation(s) involving both $q_y(\phi_g)$ and $Q_y(\phi_g)$ simultaneously). This most apparent if we appeal to the chain rule to write

$$L_g = \frac{\partial u}{\partial c_g} = \frac{\partial s_g}{\partial c_g}\frac{\partial u}{\partial s_g} + \frac{\partial s_{g+1}}{\partial c_g}\frac{\partial u}{\partial s_{g+1}} \tag{139a}$$

$$+ \frac{\partial Q_0(\phi_g)}{\partial c_g}\frac{\partial u}{\partial Q_0(\phi_g)} + \frac{\partial Q_1(\phi_g)}{\partial c_g}\frac{\partial u}{\partial Q_1(\phi_g)} \tag{139b}$$

$$+ \frac{\partial Q_0(\phi_{g+1})}{\partial c_g}\frac{\partial u}{\partial Q_0(\phi_{g+1})} + \frac{\partial Q_1(\phi_{g+1})}{\partial c_g}\frac{\partial u}{\partial Q_1(\phi_{g+1})} \tag{139c}$$

which, after some simplification, yields an expression in terms of $Q_y$:

$$L_g - L_{g-1} = \tag{140a}$$

$$\left(Q_1(\phi_{g+1}) - Q_0(\phi_{g+1})\right)\left(Q_0(\phi_{g+1})\left(V_{0\hat{\imath}} - V_{0\hat{0}}\right) + Q_1(\phi_{g+1})\left(V_{1\hat{0}} - V_{1\hat{\imath}}\right)\right) \tag{140b}$$

$$- \left(Q_1(\phi_{g-1}) - Q_0(\phi_{g-1})\right)\left(Q_0(\phi_{g-1})\left(V_{0\hat{\imath}} - V_{0\hat{0}}\right) + Q_1(\phi_{g-1})\left(V_{1\hat{0}} - V_{1\hat{\imath}}\right)\right) \tag{140c}$$

Considering that we treat arbitrary $q_y$ subject to Assumption 3, analytically solving an equation in $q$ and $Q$ simultaneously is not practical for our purposes.

## Proof of Theorem 18

**Theorem 18 Statement.** On the stable internal equilibrium hyperplane, infinitesimal perturbation of $\Phi$ by

$$\Delta_g \Phi := -\epsilon\delta(g, g+1)\left(\frac{\alpha_g}{s_1(1-s_1)}, ..., \frac{\alpha_g}{s_g(1-s_g)}, \frac{\beta_g}{s_{g+1}(1-s_{g+1})}, ..., \frac{\beta_g}{s_n(1-s_n)}\right) \tag{141a}$$

$$\alpha_g := (\mu_{g+1} + \mu_{g+2} + ... + \mu_n), \qquad \beta_g := -(\mu_1 + \mu_2 + ... + \mu_g) \tag{141b}$$

will induce motion in the system preserving $\overline{s}$ and each $\delta(h, h+1)$ for $h \neq g$. The value of $\delta(g, g+1)$ will be diminished by a ratio proportional to the **strength parameter** $\epsilon > 0$.

*Proof of Theorem 18.* For convenience, on an equilibrium hyperplane, we will write as equivalent statements

$$\left.\frac{\partial}{\partial \phi}(W_1 - W_0)\right|_{eq} = \left.\frac{\partial}{\partial \phi_g}(W_1^g - W_0^g)\right|_{eq} \tag{142}$$

We first generalize Lemma 11.1 for group-dependent feature thresholds $\phi_g$, each perturbed from $\phi_g = \phi$ at equilibrium and but applied only to the corresponding group $g$.

$$\left.\frac{\partial}{\partial \phi_g}\left(s_h[t+1] - s_h[t]\right)\right|_{eq} = \frac{1}{W_{eq}}\left.\frac{\partial}{\partial \phi}(W_1 - W_0)\right|_{eq} \begin{cases} s_g(1-s_g) & g = h \\ 0 & h \neq g \end{cases} \tag{143}$$

It follows from the definition of $\overline{s}$ that

$$\left.\frac{\partial}{\partial \phi_g}\left(\overline{s}[t+1] - \overline{s}[t]\right)\right|_{eq} = \frac{1}{W_{eq}}\left.\frac{\partial}{\partial \phi}(W_1 - W_0)\right|_{eq}\mu_g s_g(1-s_g) \tag{144}$$

and, by the definition of $\delta(h, h+1)$,

$$\left.\frac{\partial}{\partial \phi_g}\left(\delta(h, h+1)[t+1] - \delta(h, h+1)[t]\right)\right|_{eq} = \tag{145a}$$

$$\left(\frac{1}{W_{eq}}\left.\frac{\partial}{\partial \phi}(W_1 - W_0)\right|_{eq}\right)\begin{cases} s_g(1-s_g) & g = h \\ -s_g(1-s_g) & g = h+1 \\ 0 & \text{otherwise} \end{cases} \tag{145b}$$

We may now prove that perturbation of the vector $\Phi$ by the vector $\Delta_g\Phi = (\Delta_g\phi_1, \Delta_g\phi_2, ..., \Delta_g\phi_n)$ causes the system to maintain its current value of $\overline{s}$. We sum the contribution due to each $\Delta_g\phi_h$,

where $\langle \cdot, \cdot \rangle$ denotes the inner product, $\nabla_\Phi$ denotes a gradient taken with respect to the components of $\Phi$, and by linearity, $\langle \Delta_g \Phi, \nabla_\Phi \rangle$ is an operator that perturbs the system with change $\Delta_g \Phi$. Linear proportionality is denoted with $\propto$.

$$\Delta_g(\overline{s}[t+1] - \overline{s}[t])\Big|_{\text{eq}} = \langle \Delta_g \Phi, \nabla_\Phi \rangle \left( \overline{s}[t+1] - \overline{s}[t] \right)\Big|_{\text{eq}} \tag{146a}$$

$$= \sum_{h=1}^{n} (\Delta_g \phi_h) \frac{\partial}{\partial \phi_h} \left( \overline{s}[t+1] - \overline{s}[t] \right)\Big|_{\text{eq}} \tag{146b}$$

$$= \frac{-\epsilon \delta(g, g+1)}{W_{\text{eq}}} \frac{\partial}{\partial \phi} (W_1 - W_0)\Big|_{\text{eq}} \left( \sum_{h=1}^{g} \mu_h \alpha_g + \sum_{h=g+1}^{n} \mu_h \beta_g \right) \tag{146c}$$

$$\propto (-\beta_g \alpha_g + \alpha_g \beta_g) \tag{146d}$$

$$= 0 \tag{146e}$$

Next, by Eq. (145a), we consider the effect that the perturbation $\Delta_g \Phi$ has on each $\delta(h, h+1)$ at equilibrium.

$$\Delta_g(\delta(h, h+1)[t+1] - \delta(h, h+1[t]))\Big|_{\text{eq}} \tag{147a}$$

$$= \langle \Delta_g \Phi, \nabla_\Phi \rangle \left( \delta(h, h+1)[t+1] - \delta(h, h+1)[t] \right)\Big|_{\text{eq}} \tag{147b}$$

$$= \sum_{i=1}^{n} (\Delta_g \phi_i) \frac{\partial}{\partial \phi_i} \left( \delta(h, h+1)[t+1] - \delta(h, h+1)[t] \right)\Big|_{\text{eq}} \tag{147c}$$

$$= \left( \frac{-\epsilon \delta(g, g+1)}{W_{\text{eq}}} \frac{\partial}{\partial \phi} (W_1 - W_0)\Big|_{\text{eq}} \right) \begin{cases} \alpha_g & h \leq g \\ \beta_g & h > g \end{cases} \tag{147d}$$

$$- \left( \frac{-\epsilon \delta(g, g+1)}{W_{\text{eq}}} \frac{\partial}{\partial \phi} (W_1 - W_0)\Big|_{\text{eq}} \right) \begin{cases} \alpha_g & h+1 \leq g \\ \beta_g & h+1 > g \end{cases} \tag{147e}$$

$$= \left( \frac{-\epsilon \delta(g, g+1)}{W_{\text{eq}}} \frac{\partial}{\partial \phi} (W_1 - W_0)\Big|_{\text{eq}} \right) \begin{cases} \alpha_g - \beta_g & g = h \\ 0 & g \neq h \end{cases} \tag{147f}$$

Since $\alpha_g - \beta_g = 1$ by Eq. (1), We see that the discrete velocity in $\delta(g, g+1)$ induced by $\Delta_g \Phi$ is

$$-\epsilon \frac{\delta(g, g+1)}{W_{\text{eq}}} \frac{\partial}{\partial \phi} (W_1 - W_0)\Big|_{\text{eq}} \tag{148}$$

On the stable equilibrium hyperplane, where $\frac{\partial}{\partial \phi}(W_1 - W_0) > 0$ and initial discrete velocity in $\delta(g, g+1)$ is zero, the prescribed perturbation proportionately *opposes* $\delta(g, g+1)$.

# C  Additional Figures

For all settings, we display the simulated dynamics for two groups, subject to different global interventions. Streamlines approximate system time evolution. $q_0$ and $q_1$ are Gaussians with unit variance and have means $-1$ and $1$, respectively. The figures included herein are provided with little analysis and are intended to prompt further consideration for the curious reader.

## C.1  Additional Variables of Interest

In addition to the acceptance rate for Group 1 (blue; first row), we plot the false positive rate for Group 1 (red; second row) and the false negative rate for Group 1 (green; third row).

[Figure]

Figure 4: **Setting 1** (Analyzed in Section 4)

## C.2 Different $U$ and $V$

Classifier decisions will differ for a given state $\mathbf{s}$ when $V$ is modified. Similarly, the success of different strategies update with different $U$ values. The qualitative behavior of the system ultimately depends on the shape of $W_1 - W_0$ as a function of $\phi$.

[Figure]

Figure 5: **Setting 2** (Stable *and* unstable hyperplanes)

[Figure]

Figure 6: **Setting 3** (Only an unstable hyperplane; Note the negative value of $\epsilon$ for Feedback Control.)

## C.3 Different Group Sizes

[Figure]

(7a) Setting 1 with asymmetric $\boldsymbol{\mu}$: $\mu_1 = 0.7, \mu_2 = 0.3$

(7b) Setting 1 with asymmetric $\boldsymbol{\mu}$: $\mu_1 = 0.9, \mu_2 = 0.1$

## C.4 Limited Space for Acceptance

[Figure]

(8a) Setting 1, but the classifier is limited to accepting $\Pr(\hat{Y}) < 0.6$

[Figure]

(8b) Setting 1, in the classifier is limited to accepting $\Pr(\hat{Y}) < 0.3$

## C.5 Other Models

For completeness, we picture the dynamics of other models. Specifically, a Markov model like that of Zhang et al. [23] (Fig. 9) and the "best response" model of Coate and Loury [21] (Fig. 10). Note that the setting of Coate and Loury [21] assumes that agents privately know their own costs for becoming qualified, which are sampled from a group-independent distribution, rather than being uniform for all agents. We use the following set of parameters.

$$\begin{bmatrix} \mu_1 = 0.5 & \mu_2 = 0.5 \end{bmatrix}$$
$$\begin{bmatrix} T_{0,\hat{0}} = 0.2 & T_{0,\hat{1}} = 0.5 \\ T_{1,\hat{0}} = 0.1 & T_{1,\hat{1}} = 0.8 \end{bmatrix}$$
$$\begin{bmatrix} V_{0,\hat{0}} = 0.0 & V_{0,\hat{1}} = -1.0 \\ V_{1,\hat{0}} = 0.0 & V_{1,\hat{1}} = 1.3 \end{bmatrix}$$

[Figure]

Figure 9: The classifier of Setting 1, but the dynamics of Zhang et al. [23], where the probability of an agent becoming qualified in the next round given outcome $y, \hat{y}$, denoted $T_{y,\hat{y}}$, given as above. We assume $T$ is group-independent; under this assumption, disparity in qualification rates cannot persist.

[Figure]

Figure 10: The classifier of Setting 1, but the population response model of Coate and Loury [21]. The intersections of the curves shown above the phase portraits correspond to the possible fixed points of the system in qualification rate; these intersections had to be manufactured with a distribution of costs, known to agents privately, associated with qualification.