# OpenReview forum: "Unintended Selection: Persistent Qualification Rate Disparities and Interventions"
_NeurIPS.cc/2021/Conference — NeurIPS 2021 Spotlight_

### Official Review · Reviewer_R95V · 2021-07-15

**Rating:** 6
**Confidence:** 3

**Summary:**

The authors are introducing a dynamical system (agents, groups, classifiers, etc.) The agents seem to have fuzzy-like membership of the groups (randomly assigned with some prob.) and they have qualification rates for groups, etc. They have a classifier built upon a specific utility function and it is shown that it is equivalent to having a policy that works by using a simple threshold on the probability of the label conditioned on the feature (Theorem 1). This idea of fitness and replicator equations is used (Eq. 8) to build a system to analyze the stability of the dynamic system. Section 3 which covers that part is a multitude of mathematical arguments showing stability and equilibrium. Section 4 discusses what might happen if there were interventions to make the classifier follow some fairness criterion, like the equal opportunity classification. And it seems to be a direct consequence of this framework.





**Limitations And Societal Impact:**

The writing is technical but not unclear. The writing has to be technical in this case. However, some sections could still use some updates. For example, the introduction could be written such that it sets the stage to explain what is being addressed in this paper, and why it would be important and novel in more details.

The impact it would have due to the assumptions that might constraint the practical applicability of it should be addressed in a paragraph or two as well. For example, the global knowledge of the classifier is one of the assumptions that might make it impractical. In a way, the setup is just relying on the conditional distribution to make predictions. Knowing the conditional probability, is it a common assumption in this space? This needs more detailed explanations and justifications. Also, the claim that this, in practice, can be learned from enough data should be followed by some tangible example or clarification.

All in all, section 5 is good in the sense that the authors discuss their own limitations.






**Main Review:**


The new contribution is the feedback control mechanism which seems to do much better at bringing everyone to a good equilibrium compared to group based fairness measures (The Fig. 3 shows this behavior in an empirical setting as well). Section 4 seems to be interesting. It discusses what might happen if there were interventions to make the classifier follow some fairness setup. Equal opportunity classification seems to be a direct consequence of this kind of setup.

The goal is clear, but a paragraph or section on the applications/practicality in real-world settings could make the significance more pronounced.



**Time Spent Reviewing:**

3

---

> ### Author Response · Authors · 2021-08-09
> **Rebuttal to feedback from Reviewer R95V**
>
> # Summary
> We sincerely appreciate the review and the helpful feedback.
>
> The primary issue raised by Reviewer R95V is the need for the paper to consider real-world applicability. We provide this summary followed by inline responses to the review (cross-referenced using square brackets and headings).
>
> We propose to:
> 1. (in the introduction) reference real-world motivating example settings for this model, such as credit approval, school admissions, and hiring decisions. Recent revisions of this paper achieve this, along with more appropriately referencing prior work in this area (as noted in [Reviewer YFGZ.C] - heading C of our rebuttal to Reviewer YFGZ)
> 2. (During the formulation, Section 2), give concrete examples of how the reader should interpret mathematical statements: For example, a constant value of $q_y = p(X=x|Y=y)$ might be interpreted as a constant distribution of credit scores or aggregate employer scores among (non)qualified loan applicants or job applicants, respectively. A changing qualification rate $s_g = P(Y = y | G = g)$ would represent more agents in group $g$ choosing to become qualified for the loan or job.
> 3. (During the formulation, Section 2.2) referencing comment [Reviewer YFGZ.G], more concretely motivate replicator dynamics: For example, if we consider the paper-acceptance process in academia as mediated by a classifier, then a qualification strategy (e.g., only submitting high-quality papers) might "replicate" when a student adopts the (proven, successful) strategy of a collaborator or mentor.
> 4. (in Section 5) address the real-world challenges of applying our model [E].
>
> # A
> > The authors are introducing a dynamical system …
>
> To correct one potential misconception, we do not require fuzzy group membership but merely appeal to a probability distribution to allocate agents to groups with fixed and known relative frequencies. It would also be sufficient to demand that agents belong to groups with known, constant relative sizes, ignoring any probabilistic notions.
> In addition, beyond offering a model using replicator dynamics, we introduce a novel fairness intervention based on linear feedback control, as pointed out in [B].
> # B
> > The new contribution is the feedback control mechanism ...
>
> We consider the use of replicator dynamics as our primary contribution and the feedback control mechanism as a secondary contribution. The offered summary and this comment together are an appropriate characterization of our work.
> # C
> > The goal is clear, but a paragraph or section on the applications/practicality in real-world settings could make the significance more pronounced.
>
> The significance of this work may be considered an extension of existing demonstrations of the inadequacy (and potential harm) of myopic fairness interventions and the contribution of a new, dynamics-aware intervention. At present, Section 5 discusses the challenges of real-world applications, which should be supplemented with the qualifications of [Reviewer YFGZ.K].
>
> In response to this review, we also believe it appropriate to comment on the technical and ethical issues of real-world implementation, as addressed in [Summary].
> # D
> > The writing is technical but not unclear. The writing has to be technical in this case. However, some sections could still use some updates. For example, the introduction could be written such that it sets the stage to explain what is being addressed in this paper, and why it would be important and novel in more details.
>
> We appreciate the feedback. Contextualizing the contributions of this paper within existing literature is of primary importance for future revisions of the paper. We address this consideration in [Reviewer YFGZ.Summary].
> # E
> > The impact it would have due to the assumptions that might constraint the practical applicability of it should be addressed in a paragraph or two as well. For example, the global knowledge of the classifier is one of the assumptions that might make it impractical. In a way, the setup is just relying on the conditional distribution to make predictions. Knowing the conditional probability, is it a common assumption in this space? This needs more detailed explanations and justifications. Also, the claim that this, in practice, can be learned from enough data should be followed by some tangible example or clarification.
>
> By appealing to the law of large numbers, any distribution may be learned given a sufficient number of samples. This extends to the domain of machine learning and to our setting in particular. The limiting assumption simplifies our derivations, and we will be sure to address this in footnote #1 in future revisions.
>
> However, the point is well-taken that it is technically challenging to infer the posterior distribution $P(Y=y | X = x)$ given a limited amount of data.  It remains interesting to evaluate whether our dynamical model is robust to an imperfect classifier trained with a finite amount of data (or how finite data can result in different conclusions), but we leave this to future work.
>
> To further address the real-world challenges of applying our model,
> we can also expand on the technical and ethical issues of defining group boundaries and determining group membership (the practically-minded reading of lines 318-320 of the current paper). While these considerations are valid limitations of the proposed feedback mechanism, we believe they do not detract from the theoretical contributions of the paper.
>
> # F
> > All in all, section 5 is good in the sense that the authors discuss their own limitations.

---

### Official Review · Reviewer_YFGZ · 2021-07-22

**Rating:** 9
**Confidence:** 5

**Summary:**

The authors consider how qualification disparities between groups can arise without structural differences between different groups. The authors formulate the problem in terms of a classifier that interacts with groups of agents, where the agents change their strategy over time in response to the classifier, and the classifier learns new classification thresholds. The authors characterize the dynamics of this system and its fixed points, showing that there exist fixed points with unequal qualification rates between groups. The authors then discuss how imposing different fairness constraints on the classifier affect the trajectory of the system, as well as a feedback control intervention that explicitly aims to reduce qualification disparities.

**Ethical Concerns:**

A minor concern is the use of the phrase "cultural evolution", as it may suggest the notion of qualification as a "cultural" trait. At least in the US, discussion of inequality and its intersection with "culture" has been fraught, and it may be worth avoiding confusion with this. Given that the notion of "qualification strategies" is discussed in the paper, I might suggest "strategic evolution" as a potentially better phrase.

**Limitations And Societal Impact:**

This is adequate.

**Main Review:**

The paper is very well-written, and the claims are well-supported. The statements are refreshingly precise. My main concern is that the paper could do more to relate its results to existing work in the classical theory of statistical discrimination from economics and with more recent work in fair ML that has built on it.

Most pressingly, I think it is important to discuss how this model is similar or different from the Statistical Discrimination model (https://en.wikipedia.org/wiki/Statistical_discrimination_(economics)) proposed on the 70's by Phelps (https://www.jstor.org/stable/1806107) and Arrow (https://econpapers.repec.org/paper/priindrel/30a.htm, https://www.jstor.org/stable/2646963). In addition, there has been work in this line that considers the impacts of various interventions such as affirmative action (see Coate and Loury 1993 https://www.jstor.org/stable/2117558). The evolution dynamics seem quite similar between these models, so I would not be surprised if they are equivalent. I think the paper also needs to engage more with the Hu and Chen paper that was cited.

However, even if the evolutionary model here is equivalent to previous statistical discrimination models, this paper makes several contributions in terms of presenting this evolution in terms that are more accessible terms, providing a transparent analytical strategy, generalizing these models to an arbitrary number of groups, and showing how current proposals for fairness fit into their dynamics.

One angle that I think the authors could play up is the idea that the only effective intervention here is the feedback control intervention that explicitly considers the dynamics of the system, whereas current fairness metrics take a static view of the system and can interact poorly (or not at all) with the dynamics. Several papers including Liu et al 2018 Delayed Impact (in the context of a Markov model), D'Amour et al 2019 Fairness is not Static (in terms of simple simulations) and Fazelpour and Lipton 2020 Fairness from a Non-Ideal Perspective (from a more abstract philosophical point of view) have argued this point. The authors discuss some of this in related work, but the idea that there are clear unintended consequences of applying fairness criteria obliviously to dynamics is, IMO, underplayed in the current manuscript.

Some smaller concerns:

In my opinion, the motivation of replicator dynamics is quite heavy-handed, and does not characterize prior work as carefully as it could. First, this paper is not unique in assuming that there are no structural differences between groups except for this initial conditions. In addition to the statistical discrimination work cited above, the Liu et al 2018 delayed impact paper is similar (contrary to the description provided with the citation at the top of section 2.2). Secondly, it seems no less contentious to axiomatically assume replicator dynamics in this paper as it does to assume that there are incentives for investment in qualification in other work. In a social system, "fitness" will end up determined by the costs and benefits of qualification, so whether these assumptions are stated up front or tucked into the parameters chosen to characterize a system seems to make little different. I think it would make more sense to frame this work as an extension of previous work (especially if connections can be drawn to the classical statistical discrimination models) where social dynamics can lead to poor equilibria even if there are few structural disparities.

At the very least, I would recommend focusing more of the discussion 2.2 on a plausible story for the emergence of the replicator dynamics. There is a mention of "exchange of qualification strategies" before Assumption 8; elaboration on this point would be useful.

 In (16) and (20), the p-norms of D are missing exponents inside of the summation. The norm notation is also mixed up in the proof of Remark 7 in the appendix (e.g., LHS of eq 63a).

The statement of Remark 7 seems incomplete. The RHS of the iff arrow is just an expression, not a claim.

Theorem 14 should perhaps be rephrased to say that the *decision rule \hat Y* satisfies equality of opportunity under the stated assumptions.

**Time Spent Reviewing:**

3 hours

---

> ### Author Response · Authors · 2021-08-09
> **Rebuttal to feedback from Reviewer YFGZ**
>
> # Summary
> We apologize for condensing Reviewer YFGZ's excellent feedback (and our thanks for such a thoroughly helpful review) due to the character limit. We provide this summary followed by inline responses to the review (cross-referenced using square brackets and headings).
>
> We agree that our paper, as submitted, can benefit from more appropriately contextualizing its contributions in the existing literature. We have already undertaken work to correct this deficiency:
>
> The modeling innovations of this work are more appropriately framed as an extension of previous work excising structural disparities [F], such as statistical discrimination theory [B] and especially that of Coate and Loury (1993). It restates the shortcomings of myopic interventions and joins existing dynamical treatments of fairness in machine learning [C], but it is distinguished by the use of replicator dynamics, which offers advantages [D, E] (though we should address the model plausibility in more detail [G]).
>
> We appreciate the recommendation to emphasize the newly proposed feedback control mechanism as a dynamics-aware intervention [C] and acknowledge typographical mistakes that should be corrected [H, I, J].
>
> Finally, we address ethical concerns expressed by this reviewer [K] and wish to open this discussion to all reviewers.
>
> # A
> >  The authors consider ...
>
> This is a concise summary of the paper, though, as pointed out in subsequent comments by this reviewer [B, D], we are not the first to formulate the problem as characterized.
>
> Rather, our choice to use the replicator equation to model agent behavior is novel in this context [B] and is the primary contribution of this paper, while the feedback control mechanism is a secondary contribution, albeit one that we should emphasize more [C].
> # B
> > The paper is very well-written… Most pressingly… However...
>
> We agree that this paper must explain its connection to (and distinction from) the classical statistical discrimination literature and especially the Coate-Loury model. Thank you for bringing this body of work to our attention.
>
> As with the theory of statistical discrimination, disparity in our model may be derived from imperfect information [G]. Our model thus builds upon this body of prior work and extends its key insight (eschewing explicit preferences of the classifier to explain persistent discrimination).
> However, the canonical mechanism of statistical discrimination (whereby the classifier treats group membership as a statistical proxy for qualification to determine the appropriate threshold for each group - as adopted by Hu and Chen 2018) is not required in our model to endogenize persistent discrimination.
>
> The novelty of our model is highlighted by its distinction from that of Coate and Loury's, with which we share the same agent variables and model of the classifier: Coate and Loury model agents' behaviors as individually utility-maximizing (rather than collectively subject to replicator dynamics). The proportion of agents in each group that choose to become qualified in their model is, therefore, a function of only the expected marginal benefit of qualification and does not depend on the current group qualification rate, in contrast to our model.
>
> As a consequence, the Coate-Loury model requires the classifier's thresholds to be group-specific in order to realize a discriminatory equilibrium. The primary advantage of our model is its ability to model persistent discrimination even when classifier thresholds are not group-specific. Consider the ideal intervention proposed by Coate and Loury in Section II. A of the 1993 paper, which would eradicate discrimination in the Coate-Loury model but is insufficient to do so in ours:
>
> "The simplest intervention would insist that employers make color-blind assignments, requiring that B's and W's with equal test scores be treated equally. This would create equivalent investment incentives for the two groups of workers, causing them to invest at the same rate and leading employers to revise their discriminatory beliefs."
> # C
> > One angle ...
>
> We agree that we should highlight the need for responsible interventions that consider the underlying dynamics of the system (appropriately referencing prior work) and that the proposed feedback control mechanism is noteworthy as one such intervention.
> In more recent revisions of this paper, we reference prior work that explicitly considers issues with static, statistical definitions of fairness or considers dynamical treatments.
>
> * Issues with static definitions:
>   - Corbett 2018 - The measure and mismeasure of fairness
>   - (Fazelpour and Lipton 2020 is appropriate here; Thank you)
> * Markov models:
>   - Liu et al. 2018 - Delayed Impact (see [D])
>   - D'Amour et al. 2020 - Fairness is not Static
>   - Zhang et al. 2020 - How do fair decisions fare long-term
>   - Wen et al. 2019 - Fairness with Dynamics
> * Rational agent models:
>   -  Liu et al. 2020 - Disparate equilibria...
>   - (Coate and Loury 1993; Thank you)
> * Group-level model:
>   - Mouzannar et al. 2019 - From fair decision making ...
>
> # D
> > Some smaller concerns… In my opinion...
>
> This criticism of our weak characterization of existing literature and wanting justification of replicator dynamics is valid. We have improved our reference of prior work in recent revisions [C] and now more carefully emphasize that replicator dynamics has the joint advantages of
>  * endogenizing persistent disparity without unfair structural assumptions
>  * plausibly deriving from a model of local information exchange [G]
>
> Regarding our characterization of Liu et al. 2018, the structural inequality intended to be highlighted was the (erroneously interpreted) potential inequality of $\Delta \mu$ for each group in that paper, even when the classifier policies $\tau$ and score distributions $\pi$ for both groups are the same. Upon a subsequent reading, however, we see by equation (2) that this is not permitted since $\Delta(x)$ is group-independent. We agree that the inappropriate citation of Liu et al. (2018) in the first paragraph of Section 2.2 should be removed, but we still cite this work as listed in [C].
> # E
> > (paragraph of "In my opinion") Secondly...
>
> We consider a semantic distinction between local incentives (utility) and fitness:
>  * The fitness of a strategy is observed, ipso facto, as aggregated behaviors change with time, independent of mechanism.
>  * We may assume that agents actively seek to maximize utility through partially-informed, rational choices and that the resulting choices, in aggregate, recreate changes in observed behavior.
>
> There should be no functional difference in the two framings: For any (global, declarative) theory of global fitness, there ought to be a corresponding, local, imperative theory and vice-versa. The former merely provides an abstraction that is more appropriate for the focus of our work and allows us to elegantly incorporate a dependence of the future qualification rate for each group on the current qualification rate [B].
>
> Our potentially contentious, "tucked away" assumption is not replicator dynamics per se but our simplistic model of static and group-independent outcome-specific fitness values $U$ (Assumptions 9 and 10). We may imagine more sophisticated alternatives but chose to start with a simple case.
> # F
> > (paragraph of "In my opinion") I think …
>
> It is appropriate to frame this work as an extension of previous work [B], as recommended.
> # G
> > At the very least…
>
> We agree that this paper would benefit from a plausible story of the emergence of replicator dynamics under which strategies are "exchanged" between agents within groups. One such story is provided by Friedman and Sinervo (2016), and we intend to include this example in subsequent revisions of the manuscript: To quote,
> > One can also get replicator dynamics from certain sorts of imitation processes. For example, suppose that individuals are rematched pairwise at random times (with a Poisson distribution), and the player with lower payoff switches to the other player's strategy with probability proportional to its payoff advantage. It has been shown that, in the large population limit, the state obeys the replicator ODE.
>
> This picture of interaction not only yields replicator dynamics but enforces *imperfect information* between non-interacting agents (e.g., agents of different groups).
> # H
> > In (16) and (20)...
>
> Thank you. The equations will be corrected, and they preserve all subsequent results.
> # I
> > The statement of Remark 7 ...
>
> Yes, the expression on the RHS of the iff arrow in Remark 7 and (63a) should be equated to 0.
> # J
> > Theorem 14...
>
> It is perhaps clearer to rephrase Theorem 14 thus:
> >By direct consequence of Theorem 4, the classifier's policy $\pi$ automatically satisfies "equality of opportunity" for any group-independent threshold $\phi$
>
> In light of [Reviewer 2tow.D - comment D of our rebuttal to Reviewer 2tow], we will also include the definition of equality of opportunity before the theorem statement.
> # K
> > A minor concern is ... the phrase "cultural evolution"...
>
> This is, admittedly, a difficult choice since we are explicitly modeling the evolution of strategic decisions within a group by appealing to a theory of cultural transmission using memes, and this phrase reflects a conceptual distinction from prior work. On the other hand, our treatment lacks any discussion of memeplexes or the complexities of cultural diversity.
>
> We believe that we should appropriately qualify our usage of "cultural evolution," noting that our model is highly sanitized and simplified: Beyond acknowledging our assumption of "... structural and cultural homogeneity except for a single binary label." (line 316), we should note that memeplexes and cultural diversity are likely to present challenges to Asm. 9 in real-world applications.
>
> **We are open to better phrasing and welcome continued discussion on this point. We wish to open this discussion to all reviewers.**

---

> > ### Comment · Reviewer_YFGZ · 2021-08-30
> > **Thanks for the reply**
> >
> > I apologize for getting back to this late. I appreciate the reply and commitments to include clarifications of connections to previous work. My rating remains high and unchanged.
> >
> > I'd just like to emphasize two things here for revisions:
> >  - I think it would be useful to point out the substantive differences in predicted trajectories under replicator dynamics versus "rational actor" type models in statistical discrimination. How might one decide based on empirical evidence which of the frameworks to use as a model for the implications of fairness interventions?
> >
> > - Regarding the "cultural evolution" nomenclature, perhaps "group evolution" might be a more palatable phrase (my previous suggestion of "strategic evolution" fell flat to me upon revisiting). I would leave this up to the authors' judgment, but think that a useful principle might be to favor a phrase referring to a concept that is defined precisely in the paper (iirc "culture" is not, whereas "groups" are). I the authors could also stick with the current nomenclature, but would need to be ready for potential negative reactions from some readers.

---

> > > ### Author Response · Authors · 2021-09-02
> > > **We are much obliged**
> > >
> > > Thank you again for your comments. We greatly appreciate them.
> > >
> > > With respect to revisions:
> > >
> > > * We concur: addressing the differences in trajectories predicted by our model
> > > and the Coate and Loury model would provide an additional means of
> > > distinguishing the theories while also addressing potential empirical falsification.
> > > Thank you for the suggestion; it should be easy to incorporate.
> > >
> > > * Regarding the title, your principle is persuasive against using the word
> > > "Cultural". What is your opinion with respect to the title
> > >
> > >   > "Incongruent Evolution by Unconscious Selection: Persistent Qualification Disparities and Interventions"

---

> > > > ### Comment · Reviewer_YFGZ · 2021-09-03
> > > > **Sounds good!**
> > > >
> > > > The new title sounds fine to me!

---

### Official Review · Reviewer_2tow · 2021-07-26

**Rating:** 8
**Confidence:** 3

**Summary:**

The paper considers the problem of fairness. Given a binary quality, a classifier, and populations groups, how does the interaction of the evolution of the quality and the classifier accuracy evolve? The authors propose a model for this that crucially avoids tying group membership with particular attributes (like race or religion), and instead assume just that the groups are isolated. They then analyze the dynamical system and show that if there are initial disparities, they may persist forever, and finally apply fairness interventions from literature to their model.

**Limitations And Societal Impact:**

Yes

**Main Review:**

This is an excellent paper and was a pleasure to read, despite it being outside of my area. It addresses a key problem and does so crisply and elegantly. I cannot speak to the technical depth of the results (which is unfortunate for a theory paper) but the quality of the technical writing is great. I don't have many comments, except for I would add a graphic or something of the model to help the reader, and that the interventions section currently is almost incomprehensible for someone not familiar with the related work.

**Time Spent Reviewing:**

2

---

> ### Author Response · Authors · 2021-08-09
> **Rebuttal to feedback from Reviewer 2tow**
>
> # Summary
>
> We thank Reviewer 2tow for the helpful feedback.
>
> Reviewer 2tow notes the benefit that a graphical overview of our model would provide. We provide this summary followed by inline responses to the review (cross-referenced using square brackets and headings).
>
> As submitted, Equation 7 and Figure 1 detail the functional relationships between variables that define our proposed model, though the reader is required to parse additional paragraphs to interpret each symbol. We agree that a single graphic depicting these functional dependencies and deciphering the included symbols would benefit the paper. We intend to include such a graphic in the next revision of the paper.
>
> This review also highlights the need to communicate technical results without assuming familiarity with linear stability analysis and fairness interventions. Our proposal to address this, using minimal edits, is given in [B, D].
> # A
> > The paper ...
>
> We thank the reviewer for the nice summary. We only want to point out that we also introduce a new class of intervention based on feedback control.
> # B
> > This is an excellent paper and was a pleasure to read, despite it being outside of my area. It addresses a key problem and does so crisply and elegantly. I cannot speak to the technical depth of the results (which is unfortunate for a theory paper) but the quality of the technical writing is great.
>
> We appreciate the careful evaluation and the kind review!
>
> To the extent that the responsibility for highlighting the depth of technical results falls on the paper itself, we believe that paragraphs 4 and 5 of the introduction offer an appropriate summary, albeit one that requires some familiarity with linear stability analysis: We believe that the proposed changes of [D] affecting sections 1 and 3 of the paper are sufficient to bridge a lack of prior exposure.
> # C
> > I don't have many comments, except for I would add a graphic or something of the model to help the reader...
>
> Thank you for the concrete recommendation of a graphic; we address this feedback in [Summary].
> # D
> > ... [The] interventions section currently is almost incomprehensible for someone not familiar with the related work.
>
> As currently written, the interventions section (section 4) relies on the developments of section 3, which assumes prior exposure to linear stability analysis (the incorporation of which to this domain is one of the main technical contributions of this paper).
>
> We plan to improve the readability of section 3 by, e.g., explaining “perturbation” and “stability”. Instead of relying entirely on prior exposure to the subject, it is simple enough to explain that a local, linearized approximation of the dynamics can determine whether small changes to an initial state in equilibrium will result in a return to the initial state (stability) or diverge (instability).
>
> Regarding familiarity with fairness interventions, recent paper revisions feature an introduction (section 1) that succinctly introduces the concept: These interventions are constraints applied to the classifier to demand policies that result in equal (pre-specified) statistics across groups. We believe that preceding Theorems 14 and 15 with definitions of the named interventions will help to clarify this section.

---

### Author Response · Authors · 2021-08-09
**Summary of Rebuttal**

Dear Area Chairs and Reviewers,

Thank you again for providing and coordinating the reviews for our paper.
We have submitted our rebuttal to each of your reviews.
Note that we will use square brackets to denote cross-references to specific rebuttal comments, specified by reviewer ID and a letter index. Our rebuttal to each reviewer begins with a summary comment.

The first round of reviews has highlighted the need for our paper to
1. appropriately contextualize and motivate its contributions to the literature [Reviewer YFGZ.Summary]
2. address its real-world applicability and concrete interpretations [Reviewer R95V.Summary]
3. qualify its use of “cultural evolution”, a point that we wish to open to discussion [Reviewer YFGZ.K]

Additionally,
* Reviewer 2tow has noted the potential benefit of providing a graphical overview of our model. In place of Equation 7 and Figure 1, we will provide such a graphic in the next revision of the paper [Reviewer 2tow.Summary].
* We will also better motivate and explain our use of linear stability analysis [Reviewer 2tow.B, Reviewer 2tow.D] and the concept of fairness interventions [Reviewer 2tow.D].
* We adopt the minor edits outlined [Reviewer YFGZ.H, Reviewer YFGZ.I, Reviewer YFGZ.J, Reviewer YFGZ.E] to address typographical mistakes and phrasing issues.

Best,

Authors of Paper 11361

---

### Comment · Program_Chairs · 2021-10-18
**Comments on the revised paper**

Minor comments to incorporate as you're preparing the camera-ready version:

The usage of "(un)conscious" anthropomorphizes processes and policies. Consider replacing by "(un)intentional" or possibly "(un)intended". I noticed this in the title and on line 56.

The usage of "equitable" in "equitable modeling" and variants: I understand the rationale behind the choice, but it is unusual to use it to refer to a set of assumptions rather than as a description of a process/procedure or outcomes. At most places it's fine, but there are a few places where it would be better to say something more explicit and descriptive like "symmetric". A few places where I spotted this issue:

Line 60: "structurally unfair" -> "structurally asymmetric"

Line 373: "equitable setting" -> "symmetric setting"

Line 381: Not clear what's meant by "more equitable models of population response". What does it mean to be "more equitable" given that your model is already "equitable" in the sense of your operationalization? Perhaps you meant to say "more complex equitable models" or "other/further equitable models"?

Line 394: Does "inequitable modeling" refer to "modeling under inequitable/asymmetric assumptions"? Consider rephrasing to improve clarity.

A few typos:

Line 117: a typo "derivation(s) of"

Line 124: a typo, perhaps "more a" -> "a more"?

Line 361: is there a typo? should that be "F_NP” -> “F_LT"?

---

### Decision · Program_Chairs · 2021-09-27

**Decision:**

Accept (Spotlight)

**Comment:**

(The meta-review was written jointly by AC and SAC after discussion with the program chairs.)

 This paper studies how qualification disparities between groups can arise without structural differences between different groups. That is: when (optimal) machine learning systems make decisions that impact people, how can these systems cause inequality to arise and maintain inequality when it is present? Furthermore, when the system is modified to enforce standard fairness constraints, how does this impact long term inequality?

Overall, the topic is of extreme interest to the ML community (long-term fairness of systems). The reviewers found the paper to be technically clear and impactful, and all recommend acceptance, two strongly (scores = 8,9). The one reviewer only weakly recommending acceptance (score = 6) did not engage in the discussion. However, while the paper's technical contributions are crisp, several sociotechnical issues were identified by the SAC (in collaboration with ethics chairs and an additional AC).

The decision is to conditionally accept the paper. The sociotechnical issues, raised in a separate comment that serves as a sociotechnical review, must be addressed. In particular, the following changes need to be implemented for the paper to be accepted (these are copy-pasted from the sociotechnical review):

1. Review and update analogies drawn from evolutionary biology, so as to minimize the risk of essentialist misinterpretation.

2. Connect technical definitions to some real-world examples, and discuss how the assumptions made in the paper align or not with those real-world examples. Some of the key terms that require connecting to examples are "qualification" and "subpopulation". The best approach would be to introduce a plausible running example that readers can use to understand the implications of the results. Such an example would need to pin down what precisely is meant by "qualification", how the groups (subpopulations) are defined, and what the classifier is doing. If no plausible example can be produced, then the authors need to walk back their statements about the applicability of the results to real-world sociotechnical systems.

3. Be explicit about the operationalization of fairness and explain what the various assumptions and criteria around that operationalization mean in the context of real-world dynamics.

4. Discuss how the replicator dynamics might be realized in real-world examples.

5. Connect the proposed intervention to a clear motivation of mitigating harms to marginalized groups, and more explicitly clarify differences between this intervention and other approaches (such as equalized odds and demographic parity) in service of that overall goal.

UPDATE: The revisions submitted by the authors have been reviewed and the paper has been officially accepted.